# LARGER LANGUAGE MODELS ARE BETTER IN-CONTEXT LEARNERS

## ABSTRACT

We study how in-context learning (ICL) in language models is affected by semantic priors versus input–label mappings. We investigate two setups—ICL with flipped labels and ICL with semantically-unrelated labels—across various model families (GPT-3, InstructGPT, Codex, an internal model, and an instruction-tuned variant of the internal model). First, experiments on ICL with flipped labels show that overriding semantic priors is an emergent behavior of model scale. While small language models ignore flipped labels presented in-context and thus rely primarily on semantic priors from pretraining, large models override semantic priors when presented with in-context exemplars that contradict priors, despite the stronger semantic priors that larger models may hold. We next study *semantically-unrelated label ICL* (SUL-ICL), in which labels are semantically unrelated to their inputs (e.g., foo/bar instead of negative/positive), thereby forcing language models to learn the input–label mappings shown in in-context exemplars in order to perform the task. The ability to do SUL-ICL also emerges primarily with scale, and large-enough language models can even perform linear classification better than random guessing in a SUL-ICL setting. Finally, we evaluate instruction-tuned models and find that instruction tuning strengthens both the use of semantic priors and the capacity to learn input–label mappings, but more of the former.

## 1 INTRODUCTION

Language models can perform a range of downstream NLP tasks via *in-context learning* (ICL), where models are given a few exemplars of input–label pairs as part of the prompt before performing the task on an unseen example (Brown et al., 2020; OpenAI, 2023; Gemini Team, 2023, *inter alia*). To successfully perform ICL, models can (a) mostly use semantic prior knowledge to predict labels while following the format of in-context exemplars (e.g., seeing "positive sentiment" and "negative sentiment" as labels and performing sentiment analysis using prior knowledge) and/or (b) learn the input–label mappings from the presented exemplars (e.g., finding a pattern that positive reviews should be mapped to one label, and negative reviews should be mapped to a different label).

Prior work on which of these factors drives performance is mixed. For instance, although Min et al. (2022b) showed that presenting random ground truth mappings in-context does not substantially affect performance (suggesting that models primarily rely on semantic prior knowledge), other work has shown that transformers in simple settings (without language modeling pretraining) implement learning algorithms such as ridge regression and gradient descent (Akyürek et al., 2023; von Oswald et al., 2022; Dai et al., 2022).

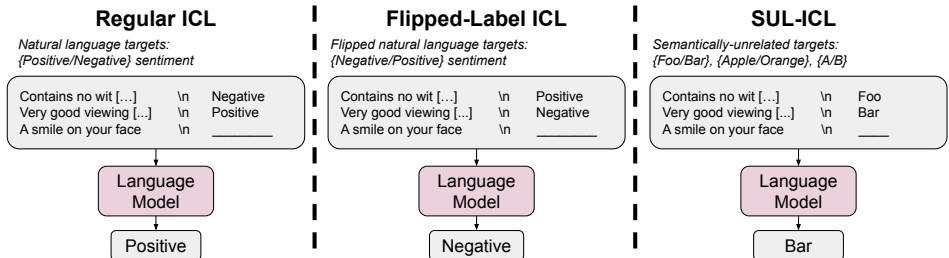

Figure 1: An overview of flipped-label ICL and semantically-unrelated label ICL (SUL-ICL), compared with regular ICL. Flipped-label ICL uses flipped targets, forcing the model override semantic priors in order to follow the in-context exemplars. SUL-ICL uses targets that are not semantically related to the task, which means that models must learn input–label mappings in order to perform the task because they can no longer rely on the semantics of natural language targets.

In this paper, we study how these two factors—semantic priors and input–label mappings—interact in several experimental settings (see Figure 1 for an example of each setting):

1. In **regular ICL**, both semantic priors and input–label mappings can allow the model to perform in-context learning successfully.
2. In **flipped-label ICL**, all labels in the exemplars are flipped, which means that semantic prior knowledge and input–label mappings disagree. Labels for the evaluation set stay the same, so for binary classification tasks, performing better than 50% accuracy in this setting means that the model is unable to override semantic priors, and performing below 50% accuracy means that the model is able to learn input–label mappings and override semantic priors.
3. In **semantically-unrelated label ICL** (SUL-ICL), the labels are semantically unrelated to the task (e.g., for sentiment analysis, we use "foo/bar" instead of "negative/positive"). Since the semantic priors from labels are removed, the model can only perform ICL by using input–label mappings.

We run experiments in these settings spanning multiple model families with varying sizes, training data, and instruction tuning (GPT-3, InstructGPT, Codex, an internal model, an instruction-tuned variant of the internal model) in order to analyze the interplay between semantic priors and input–label mappings,[1] paying special attention to how results change with respect to model scale. First, we examine flipped-label ICL, where we find that small models do not change their predictions when seeing flipped labels, but large models may flip their predictions to follow flipped exemplars (Section 3). This means that the behavior of overriding semantic priors with input–label mappings emerges with model scale, which should not be taken for granted because larger models presumably have stronger priors that are more challenging to override.

Second, we compare the SUL-ICL setting to regular ICL (Section 4). We find that small language models experience a large performance drop when semantic priors are removed, whereas large language models can perform the task well even without semantic priors from the labels. For some datasets, doing better than random in the SUL-ICL setting required substantial scaling (e.g., only the 540B internal model achieves above-random performance). We also found this to be true for high-dimensional linear classification tasks (Section 6). This means that learning input–label mappings without being given priors is also an emergent ability of large language models for those tasks.

Finally, we study the effect of instruction tuning (Min et al., 2022a; Wei et al., 2022a; Chung et al., 2022) on ICL abilities (Section 5). We find that instruction-tuned models achieve better performance than pretraining-only models on SUL-ICL settings, which means that instruction tuning increases the model's ability to learn input–label mappings. On the other hand, we also see that instruction-tuned models are more reluctant to follow flipped labels, which means that instruction tuning decreases the model's ability to override semantic priors more than it increases its ability to learn input–label mappings. Overall, our work aims to shed light on the interaction between semantic prior knowledge and input–label mappings while considering the effects of scaling and instruction tuning.

---

[1]Many factors can affect ICL, including majority-label bias and recency bias (Zhao et al., 2021). We mitigated these biases by providing equal exemplars per class and randomizing the order of input–label pairs. We studied additional factors in Appendix C.3, Appendix C.4, Appendix C.5, and Appendix C.6. It is still possible, however, that other factors could be at play, though we believe that the major factors being analyzed are the two described.

## 2 EXPERIMENTAL SETUP

### 2.1 EVALUATION TASKS

We experiment on seven NLP tasks that have been widely used in the literature (Kim, 2014; Wang et al., 2018; 2019). These evaluation tasks and an example prompt/target pair are shown in Figure 10 in the Appendix; additional dataset details are described in Appendix B. The seven tasks are: Sentiment Analysis (Socher et al., 2013, **SST-2**); Subjective/Objective Sentence Classification (Conneau & Kiela, 2018, **SUBJ**); Question Classification (Li & Roth, 2002, **TREC**); Duplicated-Question Recognition (Chen et al., 2017; Wang et al., 2018, **QQP**); Textual Entailment Recognition (Dagan et al., 2006; Wang et al., 2019, **RTE**); Financial Sentiment Analysis (Malo et al., 2014, **FP**); and Hate Speech Detection (Mollas et al., 2020, **ETHOS**).[2]

### 2.2 MODELS

We perform experiments on five language model families as shown in Table 1. We use three families of OpenAI language models accessed via the OpenAI API: GPT-3 (Brown et al., 2020), InstructGPT (Ouyang et al., 2022), and Codex (Chen et al., 2021). For GPT-3 models, ada, babbage, curie, and davinci seem to correspond to the following

| Model Family | Model Name (Abbreviation) |
|---|---|
| GPT-3 | ada (a), babbage (b), curie (c), davinci (d) |
| InstructGPT | text-ada-001 (a-1), text-babbage-001 (b-1), text-curie-001 (c-1), text-davinci-001 (d-1), text-davinci-002 (d-2) |
| Codex | code-cushman-001 (c-c-1), code-davinci-001 (c-d-1), code-davinci-002 (c-d-2) |
| Internal language model | LLM-8B, LLM-62B, LLM-540B |
| Instruction-tuned internal language model | IT-LLM-8B, IT-LLM-62B, IT-LLM-540B |

Table 1: Models used in this paper.

model sizes: 350M, 1.3B, 6.7B, and 175B (Gao et al., 2021). For InstructGPT and Codex, however, it is not publicly known what the sizes of these language models are, but we assume that they are in increasing model scale for some scaling factor.

We also experiment on three different sizes of an internal language model (LLM-8B, LLM-62B, and LLM-540B) and their instruction-tuned variants (IT-LLM-8B, IT-LLM-62B, IT-LLM-540B). Our internal language models have the same training data and protocol and only differ by model size, which provides an additional data point for the effect of scaling model size specifically. Because many experiments rely on querying OpenAI models that are not publicly-available, we do not report the compute used for these experiments.[3]

### 2.3 ADDITIONAL EXPERIMENTAL DETAILS

As additional experimental details, we follow the prior literature on in-context learning and use a different set of few-shot exemplars for each inference example (Brown et al., 2020; Chowdhery et al., 2022; Wang et al., 2023, *inter alia*). By default, we use $k = 16$ in-context exemplars per class, though we also experiment with varying number of exemplars in Section 4 and Appendix D.2. We also use the "Input/Output" template for prompts shown in Figure 10, with ablations for input format shown in Appendix C.4 and Appendix C.5, and the semantically-unrelated "Foo"/"Bar" targets as shown in Figure 10 (ablations for target type are shown in Appendix C.3). Finally, to reduce inference costs, we use 100 randomly sampled evaluation examples per dataset, as it is more beneficial to experiment with a more-diverse range of datasets and model families than it is to include more evaluation examples per dataset, and our research questions depend more on general behaviors than on small performance deltas (note that all $y$-axes in our plots go from 0%–100% accuracy).

---

[2]In preliminary experiments (Appendix C.3), we also tried two additional tasks: Question–Answering (Rajpurkar et al., 2016; Wang et al., 2018, **QNLI**) and Coreference Resolution (Levesque et al., 2012; Wang et al., 2019, **WSC**), but even the largest models had very weak performance on these tasks in many settings, so we do not include them in further experimentation.

[3]We used internal resources to evaluate our internal language models, so we do not report these numbers in order to retain anonymity.

# 3 INPUT–LABEL MAPPINGS OVERRIDE SEMANTIC PRIORS IN LARGE MODELS

To what extent do models override semantic priors from pretraining in favor of input–label mappings presented in-context? When presented in-context exemplars with flipped labels, models that override priors and learn input–label mappings in-context should experience a decrease in performance to below random guessing (assuming ground-truth evaluation labels are not flipped).

To test this, we randomly flip an increasing proportion of labels for in-context exemplars. As shown in Figure 1, for example, 100% flipped labels for the SST-2 dataset would mean that all exemplars labeled as "positive" will now be labeled as "negative," and all exemplars that were labeled as "negative" will now be labeled as "positive." Similarly, 50% flipped labels is equivalent to random labels, as we use binary classification datasets (we exclude TREC from this experiment since it has six classes). We do not change the labels of the evaluation examples, so a perfectly-accurate model that overrides priors should achieve 0% accuracy when presented with 100% flipped labels.

Figure 2 shows average model performance for each of the model families across all tasks with respect to the proportion of labels that are flipped (per-dataset results are shown in Figure 17). We see that there is a similar trend across all model families—at 0% flipped labels (i.e., no labels are changed), larger models have better performance than small models, which is expected since larger models should be more capable than smaller models. As more and more labels are flipped, however, the performance of small models remains relatively flat and often does not dip below random guessing, even when 100% of labels are flipped. Large models, on the other hand, experience performance drops to well-below random guessing (e.g,. text-davinci-002 performance drops from 90.3% with 0% flipped labels to just 22.5% with 100% flipped labels). Note that GPT-3 models remove semantic priors (i.e., perform at guessing accuracy) but does not override them (i.e., perform significantly worse than guessing), even when presented with 100% flipped labels. For this reason, we consider all GPT-3 models to be "small" models because they all behave similarly to each other this way.

These results indicate that large models override prior knowledge from pretraining with input–label mappings presented in-context. Small models, on the other hand, do not flip their predictions and thus do not override semantic priors (consistent with Min et al. (2022b)). Because this behavior of overriding prior knowledge with input–label mappings only appears in large models, we conclude that it is an emergent phenomena unlocked by model scaling (Wei et al., 2022b).

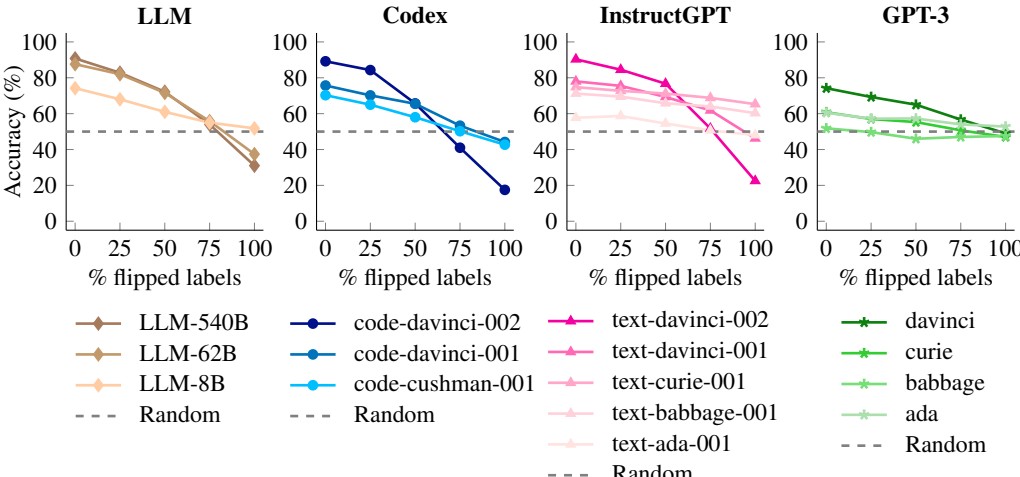

Figure 2: The behavior of overriding semantic priors when presented with flipped in-context exemplar labels emerges with model scale. Smaller models do not flip predictions to follow flipped labels (performance only decreases slightly), while larger models do (performance decreases to well below 50%). Ground truth labels for evaluation examples are not flipped, so if a model follows flipped labels, its accuracy should be below 50% when more than 50% of labels are flipped. For example, a model with 80% accuracy at 0% flipped labels will have 20% accuracy at 100% flipped labels if it flips its predictions. Accuracy is computed over 100 evaluation examples per dataset with $k = 16$ in-context exemplars per class and averaged across all datasets.

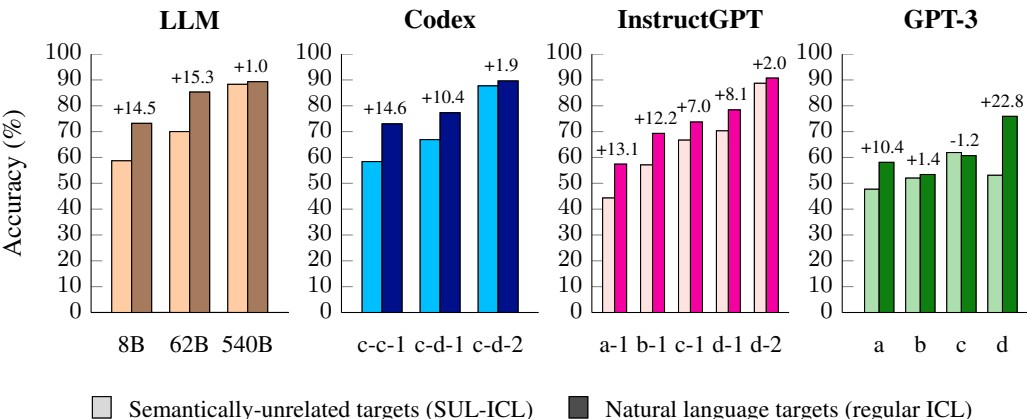

Figure 3: Small models rely more on semantic priors than large models do, as performance decreases more for small models than for large models when using semantically-unrelated targets instead of natural language targets. For each plot, models are shown in order of increasing model size (e.g., for GPT-3 models, a is smaller than b, which is smaller than c). We use $k = 16$ in-context exemplars per class, and accuracy is calculated over 100 evaluation examples per dataset and averaged across all datasets. A per-dataset version of this figure is shown in Figure 18 in the Appendix.

## 4 IN-CONTEXT LEARNING WITH SEMANTICALLY-UNRELATED LABELS CAN EMERGE WITH MODEL SCALE FOR SOME TASKS

Another way to examine how much models use semantic priors from pretraining versus input–label mappings is to replace natural language targets with semantically-unrelated targets. If a model mostly relies on semantic priors for in-context learning, then its performance should significantly decrease after this change, since it will no longer be able to use the semantic meanings of targets to make predictions. A model that learns input–label mappings in-context, on the other hand, would be able to learn these semantically-unrelated mappings and should not experience a major drop in performance.

We use an experimental setup that we call Semantically-Unrelated Label In-Context Learning (SUL-ICL) to test model behavior in these scenarios.[4] In this setup, all natural language targets are swapped with semantically-unrelated targets (we use "Foo" and "Bar" by default, although we get similar results with other semantically-unrelated targets—see Appendix C.3). For example, SUL-ICL relabels examples labeled as "negative" as "foo" and examples labeled as "positive" as "bar" for the SST-2 dataset (Figure 1). We then examine model performance in the SUL-ICL setup (in Appendix C, we investigate other aspects of the SUL-ICL setup such as remapping inputs, formatting prompts differently, changing target types, and using out-of-distribution datasets).

In Figure 3, we examine average model accuracy across all tasks on the SUL-ICL setup compared with a regular in-context learning setup (per-dataset results are shown in Figure 18). As expected, we see that increasing model scale improves performance for both regular in-context learning and SUL-ICL. The performance drop from regular ICL to SUL-ICL, however, is far more interesting. We find that using semantically-unrelated targets results in a greater performance drop from using natural language targets for small models compared with large models. Because small models are heavily affected when the semantic meaning of targets is removed, we conclude that they primarily rely on the semantic meaning of targets for in-context learning rather than learn the presented input–label mappings. Large models, on the other hand, experience very small performance drops after this change, indicating that they have the ability to learn input–label mappings in-context when the semantic nature of targets is removed.[5] Hence, the ability to learn input–label mappings in-context without being given semantic priors can also be seen as an emergent ability of model scale.

---

[4] Rong (2021) previously evaluated a setup where they replaced natural language targets with non-alphanumeric characters; our paper uses a similar setup and investigates with more-extensive experimentation.

[5] For the reasons stated in Section 3, we consider davinci to be a small model.

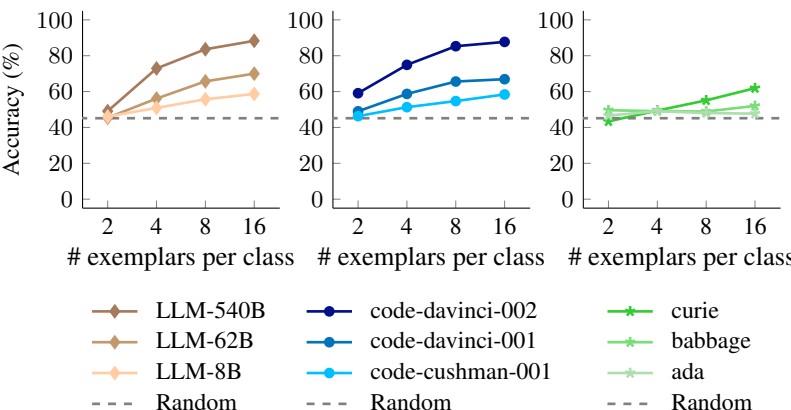

Figure 4: In the SUL-ICL setup, larger models benefit more from additional exemplars than smaller models do. Accuracy is calculated over 100 evaluation examples per dataset and averaged across all datasets. A per-dataset version of this figure is shown in Figure 19 in the Appendix.

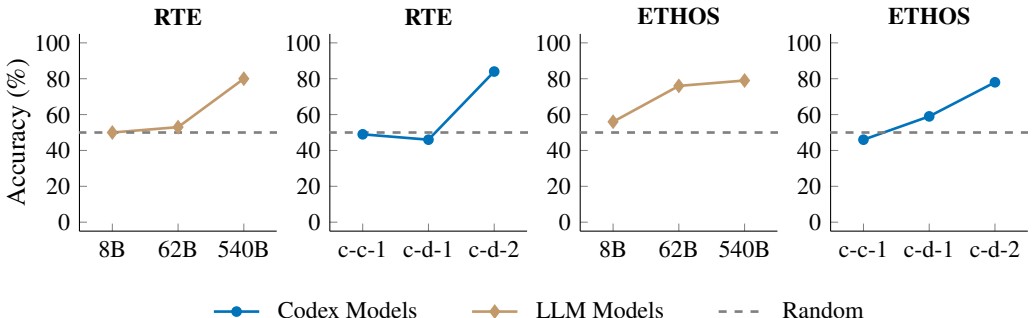

Figure 5: Some tasks in the SUL-ICL setting emerge with scale and can only be successfully performed by large-enough models. These experiments use $k = 8$ in-context exemplars per class. Accuracy is calculated over 100 evaluation examples.

We next analyze how models perform on a SUL-ICL setup when presented with an increasing number of in-context exemplars, and we show these data in Figure 4 (per-dataset results are shown in Figure 19). We find that for the three model families that we tested,[6] including more in-context exemplars results in a greater performance improvement for large models than it does for small models. This indicates that large models are better at learning from in-context exemplars than small models are, implying that large models are more capable of using the additional input–label mappings presented in context to better learn the correct relationships between inputs and labels.

Finally, looking at the per-dataset performance reveals how the ability to perform some benchmark tasks in the SUL-ICL setting emerges with scale. In Figure 5, we highlight two tasks (RTE and ETHOS) that seem particularly emergent in the SUL-ICL setting by plotting model performance at each model size for Codex and LLM models (Figure 19 shows how each model performs for each dataset). We see that performance on the RTE dataset is around random for LLM-8B and LLM-62B, yet increases to well above random for LLM-540B. Similarly, the performance on both the RTE and ETHOS datasets is around random for code-cushman-001 and code-davinci-001, then jumps to $80\%+$ for code-davinci-002. LLM models seem to emerge earlier on the ETHOS dataset, however, as the performance spikes when scaling from LLM-8B to LLM-62B. For many datasets that do not show emergence, even small models can outperform random guessing without many in-context exemplars (e.g., on SST-2, TREC, SUBJ, FP). These results show another example of how, for some tasks, the ability to learn input–label mappings in-context without being given semantic priors is only emergent in large-enough language models.

---

[6]We do not run on InstructGPT models or davinci due to the cost of running the large volume of experiments.

# 5 INSTRUCTION TUNING WITH EXEMPLARS IMPROVES INPUT–LABEL MAPPINGS LEARNING AND STRENGTHENS SEMANTIC PRIORS

A popular technique for improving the performance of pretrained language models is to finetune them on a collection of NLP tasks phrased as instructions, with few-shot exemplars as part of the finetuning inputs (Min et al., 2022a; Wei et al., 2022a; Chung et al., 2022; Longpre et al., 2023). Since instruction tuning uses natural language targets, however, an open question is whether it improves the ability to learn input–label mappings in-context or whether it strengthens the ability to recognize and apply semantic priors, as both would lead to an improvement in performance on standard ICL tasks.

To study this, we run the same experiments from Section 3 and Section 4, and we now compare LLM models to their instruction-tuned versions (IT-LLM). We do not compare InstructGPT against GPT-3 models in this experiment because we cannot determine if the only difference between these model families is instruction tuning (e.g., we do not even know if the base models are the same).

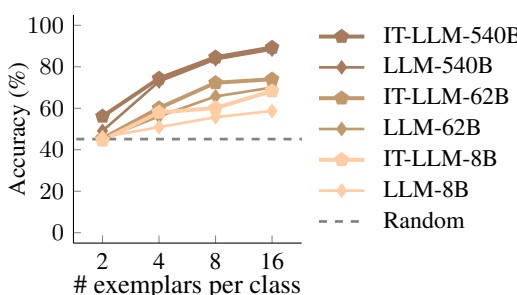

Figure 6: Instruction-tuned language models are better at learning input–label mappings in the SUL-ICL setting than pretraining-only language models are. Accuracy is calculated using 100 evaluation examples per dataset and averaged across six datasets. A per-dataset version of this figure is shown in Figure 20 in the Appendix.

Figure 6 shows the average model performance across all datasets with respect to the number of in-context exemplars for LLM and IT-LLM models. We see that IT-LLM performs better in the SUL-ICL setting than LLM does, an effect that is most prominent in small models, as IT-LLM-8B outperforms LLM-8B by 9.6%, almost catching up to LLM-62B. This trend suggests that instruction tuning strengthens the ability to learn input–label mappings (an expected outcome).

In Figure 7, we show model performance with respect to the proportion of labels that are flipped for each LLM and IT-LLM model. We find that, compared to pretraining-only models, instruction-tuned models are worse at flipping their predictions—IT-LLM models were unable to override their semantics more than what could be achieved by random guessing, even with 100% flipped labels. Standard LLM models, on the other hand, could achieve as low as 31% accuracy when presented with 100% flipped labels. These results indicate that instruction tuning either increases the extent to which models rely on semantic priors when they are available or gives models more semantic priors, as instruction-tuned models are less capable of flipping their natural language targets to follow the flipped labels that were presented. Combined with the result from Figure 6, we conclude that although instruction tuning improves the ability to learn input–label mappings, it concurrently strengthens the usage of semantic priors, similar to the findings in Min et al. (2022a).

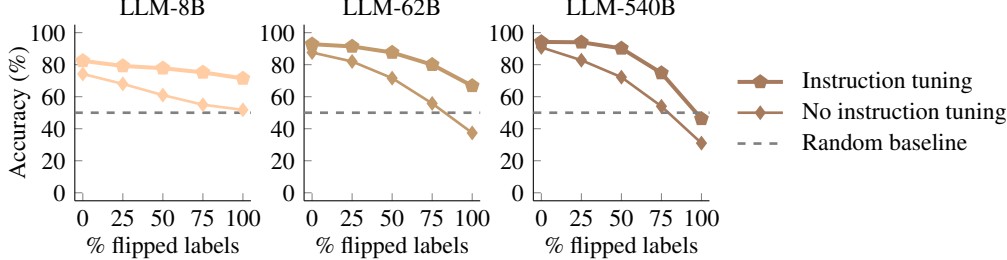

Figure 7: Instruction-tuned models are worse than pretraining-only models are at learning to override semantic priors when presented with flipped labels in-context. We use $k = 16$ in-context exemplars per class, and accuracy is calculated using 100 evaluation examples per dataset and averaged across six datasets. A per-dataset version of this figure is shown in Figure 21 in the Appendix.

# 6 LARGE LANGUAGE MODELS CAN PERFORM LINEAR CLASSIFICATION

In addition to the natural language reasoning abilities that we studied throughout the rest of the paper, we also seek to learn about how model scale affects the ability to perform other tasks. Specifically, we look at the linear classification task, where large models should perform better than small models (especially at high dimensions) if their greater capacity to learn input–label mappings as shown in Section 4 also holds for non-natural-language tasks.

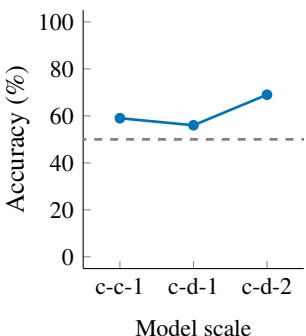

To analyze this, we create $N$-dimensional linear classification datasets and examine model behavior with respect to the number of dimensions in the SUL-ICL setup. In these datasets, we provide $k$ $N$-dimensional points above a threshold and $k$ $N$-dimensional points below that same threshold as in-context exemplars, and the model must determine whether an $N$-dimensional evaluation point is above or below the threshold (we do not tell the model the equation or the threshold). When selecting random $N$-dimensional points, we use random integers between 1 and 1000 for each coordinate value. Algorithm 1 in the Appendix shows the precise dataset generation procedure.

Figure 8: Successfully performing 16-dimensional linear classification emerges with model scale for Codex models. Accuracy is calculated over 100 evaluation examples with $k = 16$ in-context exemplars per class. Per-dimension results are shown in Figure 9 in the Appendix.

In Figure 8, we show Codex model performance on $N = 16$ dimensional linear classification (per-dimension results on Codex and LLM models are shown in Figure 9 in the Appendix). The largest model outperforms random guessing by 19% on this task, while smaller models cannot outperform random guessing by more than 9%, suggesting that there exists some scaling factor that allows large-enough language models to perform high-dimensional linear classification.

In Figure 9, we show model performance for Codex and LLM models versus an exponentially-increasing number of dimensions $N$ (the data generation procedure is shown in Algorithm 1). We also include results from a standard polynomial SVM implemented via scikit-learn (`svm.SVC(kernel='poly')`) for comparison. We find that for the Codex model family, the largest model can successfully perform linear classification up to $N = 64$, while the smaller models reach guessing performance at approximately $N = 16$. For LLM models, on the other hand, model scale does not seem to significantly correlate with the number of dimensions to which the model can

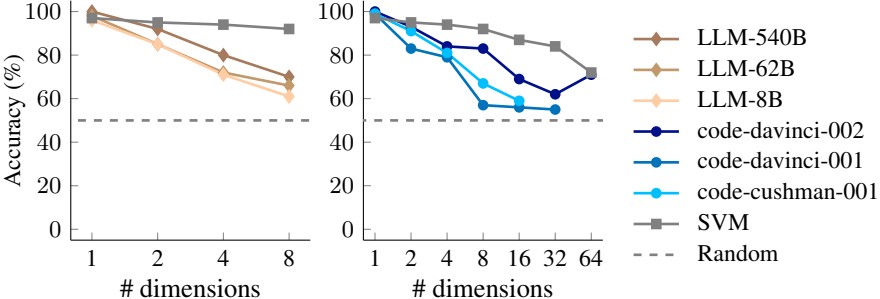

Figure 9: The largest Codex model (code-davinci-002) can perform linear classification up to 64 dimensions, while smaller Codex models do not outperform random guessing at 16 dimensions. LLM models can all perform linear classification up to 8 dimensions with little difference in performance with respect to model scale. Standard SVM algorithm performance shown for comparison. Accuracy is calculated over 100 evaluation examples per dataset with $k = 16$ in-context exemplars per class.

perform linear classification, though all LLM models can perform linear classification up to at least $N = 8$.[7] Neither LLM models nor Codex models can outperform an SVM baseline.

These results suggest that model size alone does not necessarily unlock the ability to perform linear classification at high dimensionality (since LLM-540B does not outperform LLM-8B or LLM-62B), but instead imply that there is another scaling factor seen in the Codex models that allows this ability to emerge. Because we do not know the particular scaling factors of the Codex model family, we leave exploration as to what factors unlock this ability to future work.

## 7  RELATED WORK

### 7.1  IN-CONTEXT DEMONSTRATIONS PROVIDE SEMANTIC PRIOR KNOWLEDGE

There has been a growing body of work on in-context learning that suggests that good performance is primarily driven by semantic priors and other factors such formatting and inducing intermediate token generation. For instance, Min et al. (2022b) showed the surprising result that using random ground-truth labels in exemplars barely hurts performance, suggesting that performance is instead mainly driven by the label space, distribution of input text, and overall format of the sequence. Along the same lines, Madaan & Yazdanbakhsh (2022) and Wang et al. (2022) show that for chain-of-thought prompting (Wei et al., 2022c), logically-incorrect prompts do not hurt performance on multi-step reasoning tasks. On a theoretical level, Xie et al. (2022) provide an explanation of in-context learning in which transformers infer tasks from exemplars because they are trained to infer latent concepts during pretraining, and prior knowledge obtained from pretraining data can then be applied to in-context examples. Finally, Reynolds & McDonell (2021) showed that clever zero-shot prompts can outperform few-shot prompts, which implies that some NLP tasks benefit more from leveraging the model's existing knowledge than from learning about the task from in-context exemplars. In this paper, we do not contest the claim that language models can benefit greatly from semantic prior knowledge—our results instead add nuance to the understanding of ICL by showing that, when semantic prior knowledge is not available, large-enough language models can still do ICL using input–label mappings. Our experiments are consistent with Min et al. (2022b) for models scaling up to davinci, and we show that learning input–label mappings only emerges with larger models (e.g., LLM-540B, text-davinci-002, and code-davinci-002).

### 7.2  LEARNING INPUT–LABEL MAPPINGS

Other recent work has suggested to some degree that language models can actually learn input–label mappings from exemplars given in-context, which is a more-attractive ability than using semantic priors because it means that the model would be able to perform a wide range of tasks even if those tasks are not seen in or even contradict pretraining data. For instance, transformers trained from scratch can perform in-context learning on linear-regression datasets with performance that is comparable to the least-squares estimator (Garg et al., 2022), and recent work has shown that transformers can do so by implementing standard learning algorithms such as ridge regression and gradient descent (Akyürek et al., 2023; von Oswald et al., 2022; Dai et al., 2022). In the natural language setting, Webson & Pavlick (2022) showed that language models learn just as fast with irrelevant or misleading prompts during finetuning or prompt-tuning. Our work makes similar claims about the ability for language models to learn tasks via input–label mappings only, though it differs crucially in that we observe frozen pretrained transformers without any additional learning. Additionally, our work focuses on empirically demonstrating that larger language models are better at learning input–label mappings in-context. Related work from Shi et al. (2024) provides theoretical explanations for this phenomenon, proposing that when performing in-context learning, larger language models cover more hidden features, whereas smaller language models emphasize important hidden features.

---

[7]We do not experiment with $N > 64$, $N > 32$, and $N > 16$ for code-davinci-002, code-davinci-001 and code-davinci-002, respectively, because of context length constraints. We do not experiment with $N > 8$ for LLM models for the same reason.

### 7.3 EMERGENT PHENOMENA IN LARGE LANGUAGE MODELS

In this paper we have also focused on the effect of scaling on in-context learning, which relates to a nascent body of work showing that scaling language models leads to qualitatively-different behavior (Ganguli et al., 2022; Wei et al., 2022b; Srivastava et al., 2022). For instance, it has recently been shown that scaling up language models can allow them to perform a variety of challenging tasks that require reasoning (Wei et al., 2022c; Chowdhery et al., 2022; Kojima et al., 2022; Zhou et al., 2023). Our experimental findings on the flipped-label ICL setup show that language models can learn input–label mappings even when the input–label mapping contradicts the semantic meaning of the label, demonstrating another type of symbolic reasoning where language models can learn input–label mappings regardless of the actual identity of the labels. Although we have shown that this behavior is emergent with respect to model scale, the investigation of why scaling unlocks such behaviors (Xie et al., 2022; Chan et al., 2022) is still an open question that we leave for future work.

## 8 LIMITATIONS

While our study sheds light on the interplay between semantic priors and input–label mappings in in-context learning for language models, there are several limitations to our work. An open question is how to apply our findings in a generative setting—we evaluated models on a range of classification tasks with discrete labels, but we did not test any generation tasks since it is unclear how to study the role of in-context demonstrations in those settings. Additionally, we examined the emergent ability of large language models to override semantic priors and learn input–label mappings. It is unknown, however, whether these emergent abilities may be affected by changes to the pretraining objective, architecture, or training process, and future work could investigate these factors. Moreover, as stated in Section 2.3, our experiments were conducted using only 100 evaluation examples per dataset because we prioritized using more datasets and model families over more evaluation examples per dataset. Future work could thus evaluate models on our settings using larger evaluation sizes per dataset. While we prioritized evaluating more model families, we note that our experiments in Section 5 were only conducted on LLM models, leaving open the question of whether the result generalizes to other model families as well.

## 9 CONCLUSIONS

In this paper, we examined the extent to which language models learn in-context by utilizing prior knowledge learned during pretraining versus input–label mappings presented in-context. We first showed that large language models may override semantic priors when presented with enough flipped labels (i.e., input–label mappings that contradict prior knowledge), and that this behavior emerges with model scale. We then created an experimental setup that we call Semantically-Unrelated Label In-Context Learning (SUL-ICL) which removes semantic meaning from labels by replacing natural language targets with semantically-unrelated targets. Successfully doing ICL in the SUL-ICL setup is another emergent ability of model scale. Additionally, we analyzed instruction-tuned language models and found that instruction tuning improves the capacity to learn input–label mappings but also strengthens semantic priors. Finally, we examined language model performance on linear classification tasks, finding that successfully performing high-dimensional linear classification emerges with model scale. These results underscore how the in-context learning behavior of language models can change depending on the scale of the language model, and that larger language models have an emergent ability to map inputs to many types of labels, a form of true symbolic reasoning in which input–label mappings can be learned for arbitrary symbols.

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

# Appendix

## Table of Contents

# A  FREQUENTLY ASKED QUESTIONS

## A.1  IS FOLLOWING FLIPPED LABELS A DESIRABLE OR UNDESIRABLE BEHAVIOR?

While overriding semantic priors in favor of input–label mappings shown in-context is not inherently a positive behavior, there are some reasons why it may still be a desired behavior in language models. First, while memorizing information is important, being able to manipulate existing knowledge to learn and adapt to new information is a crucial feature of intelligence (Newell, 1980; Santoro et al., 2021). Humans, for example, have broad knowledge of what words mean but are able to learn new patterns from only a few examples. Hence, humans would be able to realize that labels are flipped and answer accordingly. Second, a useful language model that can adapt to new information should be able to update its knowledge given new information in-context. It should also prioritize in-context information (which could be, for example, more recent) over prior knowledge (which could be outdated). This ability would be practically useful in many applications. As an example, a language model trained on knowledge before 2023 should be able to override some prior knowledge if new information is presented in-context that is more up-to-date. Third, being able to override priors is important since it could help show that language models do not memorize but rather are able to manipulate symbols regardless of the identity of those symbols. This would be a way to demonstrate the ability to learn symbols, related to Fodor & Pylyshyn (1988) and Smolensky (1988).

At the same time, however, this ability to follow flipped labels can also demonstrate some fragility in how language models perform in-context learning. Our findings show that large-enough language models may actually prefer input–label mappings presented in-context to the point of overriding their prior knowledge from pretraining. This could suggest that large-enough models may be more susceptible to adversarial prompts that contain untrue or dangerous in-context information, similar to how larger language models are more sycophant than smaller language models (Perez et al., 2022).

## A.2  WHY ARE LARGER MODELS BETTER AT IN-CONTEXT LEARNING?

In Section 3, we demonstrated the result that larger language models are better at following flipped labels presented in-context than smaller models are. This result is striking since larger models should presumably have stronger priors that are more challenging to override. While it is impossible to know exactly why this behavior occurs, it could be a result of (a) larger models preferring input–label mappings presented in-context over prior knowledge or (b) larger models being more sample efficient (Kaplan et al., 2020; Chowdhery et al., 2022) and more-effectively utilizing the in-context exemplars than smaller models. (Shi et al., 2024) also proposes that when performing in-context learning, large language models cover more hidden features, whereas smaller language models emphasize important hidden features.

One way in which future work could gain insight into why larger models are better at in-context learning could be to study models using mechanistic interpretability. For example, analyzing attention and activation patterns could reveal how models of varying sizes process in-context examples. Furthermore, circuit analysis could identify specific mechanisms that might allow larger models to utilize new information over existing priors. Further analysis in this direction could potentially provide insight on how the ability to override priors is implemented and what structural changes enable stronger in-context learning capabilities.

## A.3  WHY ARE ALL GPT-3 MODELS CONSIDERED TO BE "SMALL" IN THIS PAPER?

In Section 3 and Section 4, we saw that GPT-3 models behaved similarly to small models from the other model families. As another example, LLM-62B outperforms the largest GPT-3 model (Brown et al., 2020) by 4.85% on SuperGLUE (Wang et al., 2019), despite being approximately three times smaller. Because GPT-3 models perform similarly to small models from other families, we view them as being "small." One possible explanation for this behavior is that GPT-3 was trained on less data and lacked many modern architectural and data improvements compared to newer language models such as the tested LLM, Codex (Chen et al., 2021), and GPT-3.5 (Ouyang et al., 2022) models. It is thus not entirely unexpected that GPT-3 models behave like smaller models from other families.

### A.4 HOW CAN THESE EMERGENT ABILITIES HELP INSPIRE FUTURE ALGORITHMS?

We observed in Section 4 that only large-enough models can do in-context learning in the SUL-ICL setup. Smaller models, however, are unable to do so. This raises the question of how to better improve this ability during pre-training. For example, including data during pretraining that forces the model to learn rule-based correlations (e.g., code) may improve the resulting model's in-context learning abilities (which may be consistent with the Codex models' strong in-context learning abilities shown in Section 3, Section 4, and Section 6). Another possibility is to change the transformer architecture to assign additional attention weight to input–label relationships given in-context, which could help smaller models perform in-context learning more effectively.

### A.5 WOULD THESE FINDINGS TRANSLATE TO GENERATIVE TASKS?

Because our in-context learning settings required discrete labels, we only experimented on classification-type tasks. Additional evaluations on generative tasks would help demonstrate how models behave outside of classification tasks, though a necessary consideration is how to best apply our setups in a generative setting. For example, it is unclear how to flip the labels of a generation-type task or how to best evaluate a model's response for correctness in a generative setting. One possibility could be to insert facts that differ from a model's semantic priors in the prompt and measure whether the model generates a fact that matches its prior knowledge or that matches the facts injected into the prompt. We did not investigate this, however, because of the difficulty of evaluating responses in a generative setting, and because it is unclear if models that have not been instruction tuned would understand that they should attempt to learn from the injected facts.

### A.6 IS THERE SOMETHING UNIQUE IN THE CODE THAT CAUSES THESE RESULTS?

We've released anonymized code for implementing our basic evaluation pipeline from NLP dataset retrieval to evaluating OpenAI models at https://anonymous.4open.science/r/in-context-learning-E3BC. Appendix E also contains full prompt examples for each dataset that would allow one to reproduce the experimental settings from our work. To our knowledge, there are no confounding factors in the code that affect the results obtained in our experiments.

### A.7 DOES IN-CONTEXT LEARNING BEHAVIOR CHANGE WITH RESPECT TO SCALING FACTORS OTHER THAN MODEL SIZE?

In our work, we investigated several model families to study how in-context learning behaviors differ across model scale. Namely, each model in the LLM model family is trained on the same training data and uses the same training protocol; the only difference is model size. This means that our findings on the LLM and IT-LLM models show the effect of purely scaling model parameter count. On the other hand, we also used GPT-3, InstructGPT, and Codex models; for these families, it is not publicly known if the models within each family are only scaling in terms of parameter count. It is likely that there are other scaling factors (e.g., better training data, a better model architecture) that increase the scale of that model. For example, code-davinci-001 and code-davinci-002 may actually have the same number of parameters, but code-davinci-002 could be trained on better data. In our paper, we assume that for these model families, the models are increasing in scale for some scaling factor, not just parameter count.

Our experiments show that in-context learning behavior often changes with respect to model scale for all model families. This means that (a) purely scaling parameter count can result in the behavior of overriding semantic priors and (b) models that are larger for scaling factors other than parameter count (e.g., quality of training data) still exhibit the behavior of being able to override semantic priors when performing in-context learning. We can see (a) because the behavior exists in the LLM model family, for which models only differ by parameter count. We can see (b) because the behavior exists in the GPT-3, InstructGPT, and Codex model families, for which models do not necessarily only differ by parameter count (although we cannot confirm which exact factors are contributing because this information is not publicly known).

### A.8 How can one control in-context learning behaviors?

The behavior of overriding semantic priors can be both beneficial (e.g., teaching the model an updated fact) and harmful (leaving the model susceptible to false knowledge in a prompt). An open question is thus how one might be able to control the behavior of overriding semantic priors in language models.

One example of a harmful side effect of language model's susceptibility to override priors when presented with in-context examples is many-shot jailbreaking (Anil et al., 2024), where a large number of in-context examples of unsafe dialogues are presented to a language model in order to override its prior safety training. Anil et al. (2024) found that a useful intervention to prevent this strategy of jailbreaking is to finetune the language model on examples where the model follows its prior knowledge and ignores demonstrations given in-context. These findings suggest that some control over the extent to which large language models override priors with in-context examples can be gained via supervised finetuning. For example, if one desires a large language model that always conforms to its prior knowledge, supervised finetuning on examples where the final answer is independent of the few-shot examples may reduce the model's tendency to override prior knowledge when shown in-context examples.

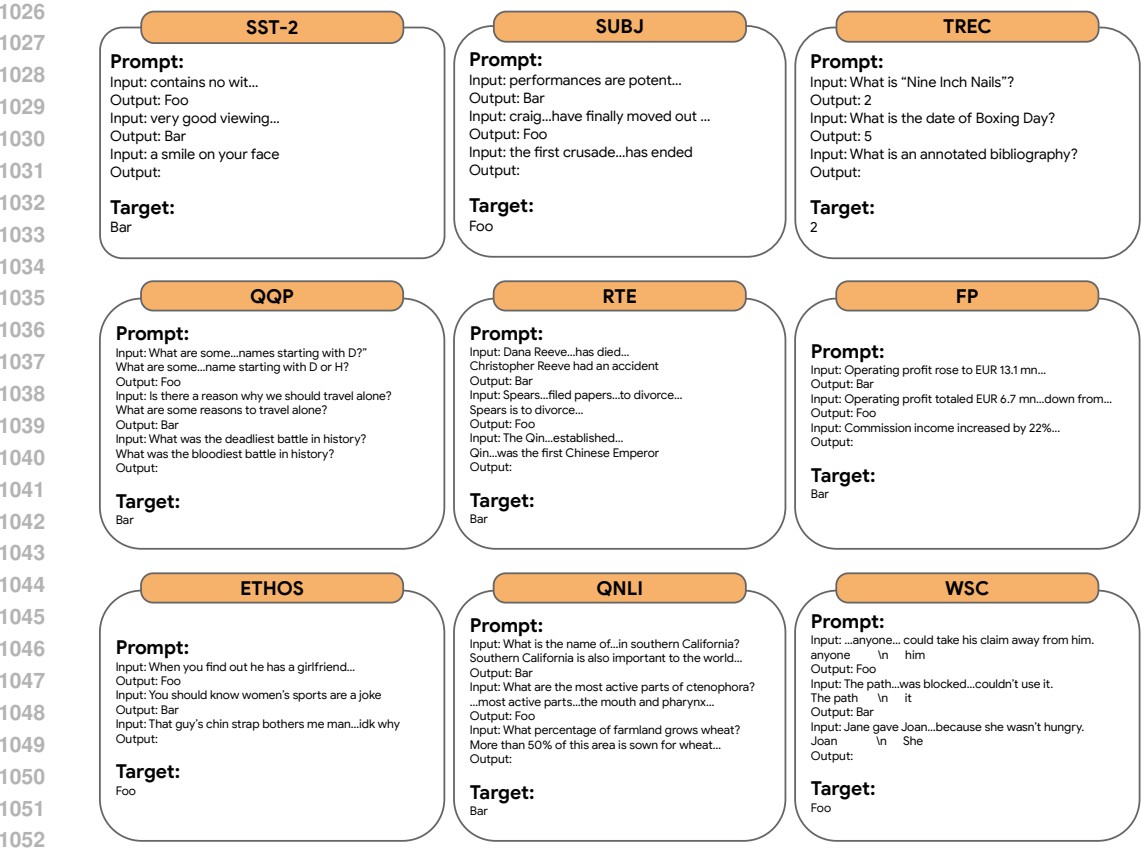

Figure 10: Prompt formatting for all datasets. We use varying number of in-context exemplars per class in our experiments, but we show one in-context exemplar per class in this figure for conciseness.

## B  DATASET CREATION

Figure 10 shows example prompts with inputs and targets from each dataset that we tested (full prompt examples for the seven datasets used in the main paper are shown in Appendix E). For each natural language task, we use the version of the dataset that is available on HuggingFace (Lhoest et al., 2021), and we randomly choose in-context exemplars from the training set and evaluation examples from the validation set, following Min et al. (2022b). For datasets without existing train/validation splits, we use a random 80/20 train/validation split.

For the FP dataset, we use the `sentences_allagree` subset. We also use the `binary` subset of the ETHOS dataset. Additionally, we use the six coarse labels for the TREC dataset.

## C  INVESTIGATING THE SUL-ICL SETUP

### C.1  SUL-ICL IS EASIER THAN FLIPPED-LABEL ICL

A natural question about the SUL-ICL setup is whether it is more difficult than the flipped labels setup. Intuitively, one would expect that the SUL-ICL setting is easier than the flipped-label setting because while the model needs to override contradiction labels in the flipped-label setting, it does not need to do so in the SUL-ICL setting.

We investigate this question by analyzing model outputs in the SUL-ICL and flipped-label settings. We use the same results from Section 4 to show model performance in the SUL-ICL setting (specifically, we use the per-dataset results from Figure 3). For the flipped-label setting, we use model outputs and

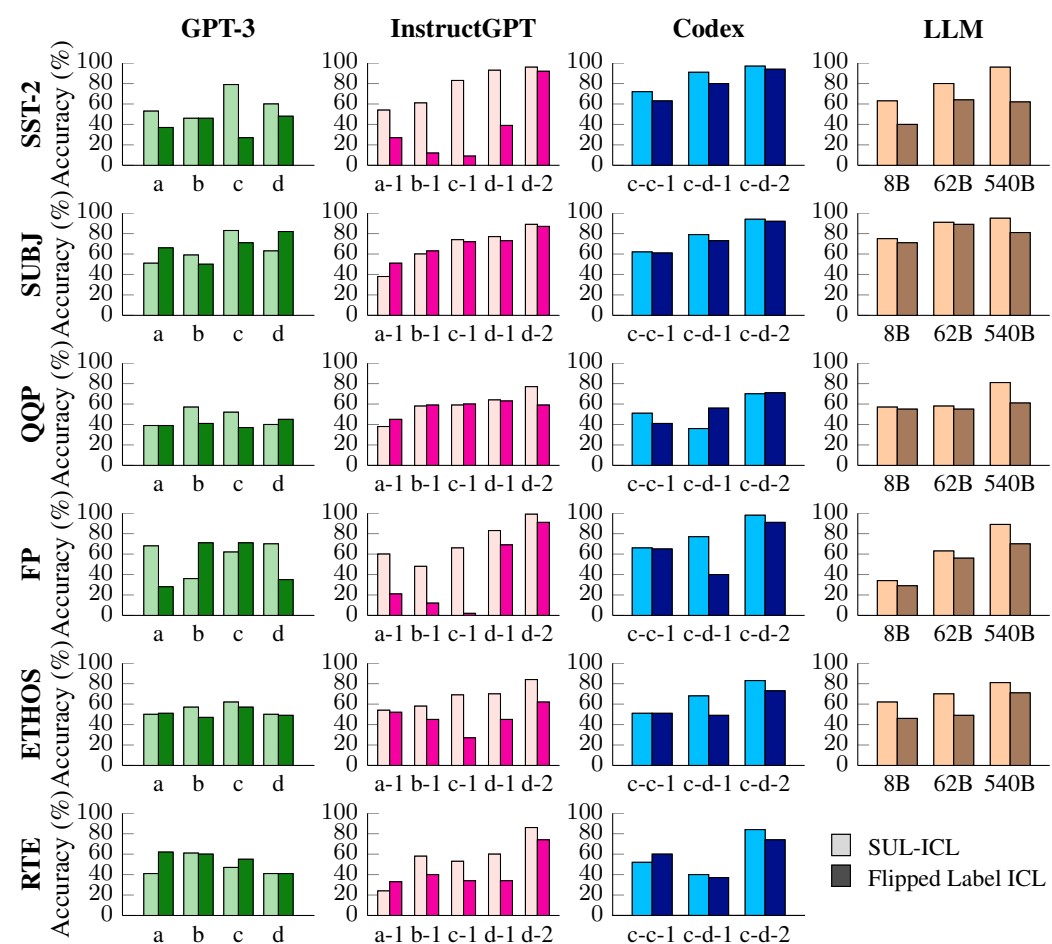

Figure 11: Models perform better in the SUL-ICL setting than they do in the flipped-label setting. Accuracy calculated over 100 evaluation examples with $k = 16$ in-context exemplars per class.

evaluation examples with 100% flipped labels (see Section 3), and we then flip evaluation examples (i.e., higher accuracy means the model can follow flipped predictions) to make comparison easier.[8]

In Figure 11, we compare model performance in the SUL-ICL setting with model performance in the flipped-label setting. We find that performance is almost always higher in the SUL-ICL setting than it is in the flipped-label setting. In particular, medium-sized models perform much worse in the flipped-label setting than they do in the SUL-ICL setting, with performance differing by up to 74% (text-curie-001 on SST-2). Small and large models, on the other hand, see smaller but still significant performance drops when using flipped-labels compared to SUL-ICL labels.

These results suggest that the SUL-ICL setting is indeed easier than the flipped-label setting, and that this trend is particularly true for medium-sized models. Small and large models are still affected by the setting, though perhaps to a lesser degree because small models often do not outperform guessing anyway and large models are more capable of overriding semantic priors (i.e., perform better in flipped-label settings). This may be an indication that the flipped-label setting's requirement of overriding priors is more difficult than learning mappings to semantically-unrelated labels.

---

[8]The accuracy shown in this section is not always equivalent to 100% minus the accuracy shown in Section 3 because models, particularly small ones, will occasionally return a prediction that is not one of the inputted labels (e.g., trying to answer a question in QQP instead of labeling questions as duplicate/non-duplicate).

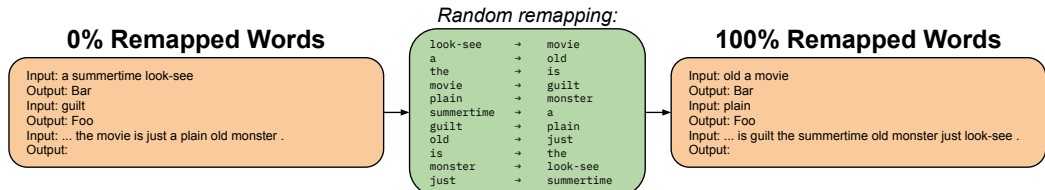

Figure 12: An overview of remapped inputs, where words are remapped to other words to reduce the semantic meaningfulness of inputs. We use prompts with $k = 16$ in-context exemplars per class in our experiments, but we show $k = 1$ in-context exemplar per class in this figure for conciseness.

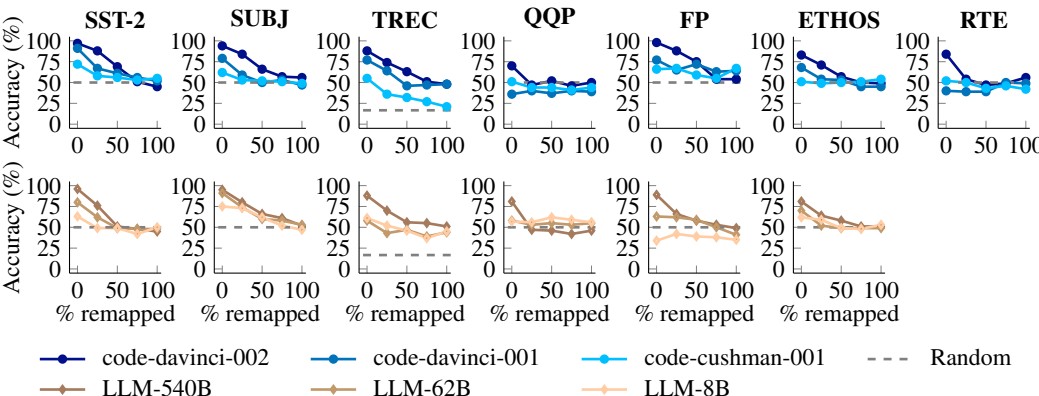

Figure 13: Language models fail in the SUL-ICL setting when input words are remapped. Accuracy is calculated over 100 evaluation examples with $k = 16$ in-context exemplars per class.

## C.2 REMAPPING INPUTS HURTS PERFORMANCE

As a sanity check, we want to show that even large models cannot succeed in the SUL-ICL setup in all environments. For example, when presented with semantically-meaningless inputs, even the largest models should not be able to perform the task because there are no longer any semantics that can be used to learn what the task is (the SUL-ICL setup already removes semantics from labels).

To show this, we remap an increasing percentage of input words to other input words at a per-prompt level. We first compile the set of all words used in the inputs for a given prompt, and we then map a randomly selected proportion of those words to other randomly selected words, thereby reducing the semantic meaningfulness of inputs. In this setup, 0% remapped words means that no input words have been changed (i.e., regular SUL-ICL), and 100% remapped words means that every input word has been remapped (i.e., inputs are now a concatenation of random words from other inputs, making them essentially meaningless). An example of this procedure is shown in Figure 12.

In Figure 13, we show model performance with respect to the proportion of remapped words. We find that small models generally approach guessing performance at 25%–50% remapped words, while large models see linear performance drops, usually reaching guessing accuracy at 75%–100% remapped words. At 100% remapped input words, even the largest models (code-davinci-002 and LLM-540B) are unable to beat random guessing on almost all datasets.[9]

These results suggest that larger models are more robust to input noise, but only to some extent because they still cannot consistently learning the required mappings to unscramble the words when a large enough proportion of words have been remapped. Indeed, 100% remapped words is most likely too difficult of a task to learn for these models, as the only way to solve the task reliably would be to unscramble most mapped words back to their original words, which would be difficult for even a human to do given the large number of input words per prompt.

---

[9]TREC is the exception, though it is unclear why large models can outperform random guessing on TREC given that 100% remapped input words is equivalent to completely-scrambled inputs.

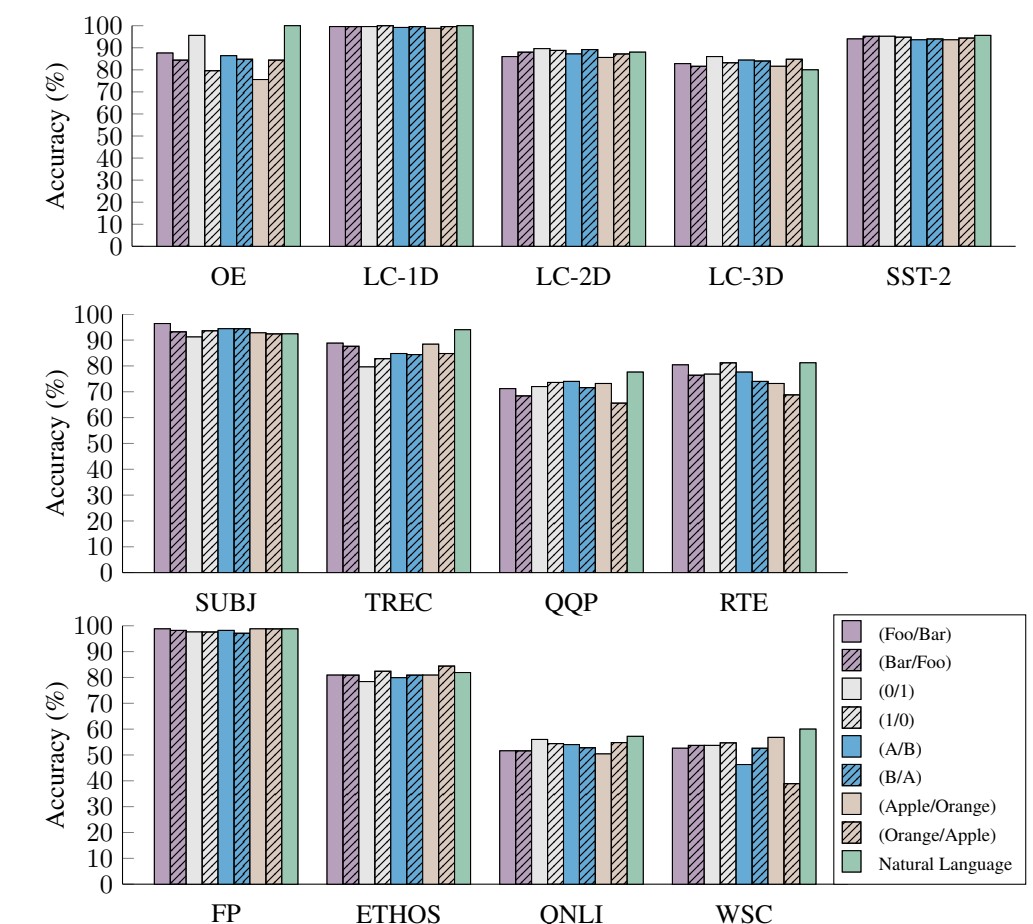

Figure 14: SUL-ICL works with many types of semantically-unrelated targets. All tasks are binary classification except TREC, which is six-way classification and uses (Foo/Bar/Iff/Roc/Ket/Dal), (0/1/2/3/4/5/6), (A/B/C/D/E/F), and (Apple/Orange/Banana/Peach/Cherry/Kiwi). Reversed targets such as (0/1) and (1/0) means that, for example, if (0/1) assigns 0 = negative and 1 = positive for sentiment analysis, then (1/0) assigns 1 = negative and 0 = positive. "Natural language" indicates that natural language targets are used (i.e., regular ICL). Accuracy is calculated over 250 evaluation examples inputted to code-davinci-002 with $k = 16$ in-context exemplars per class.

## C.3 MANY TARGET TYPES WORK

In Section 4, we showed that large language models can learn input–label mappings for one set of semantically-unrelated targets ("Foo" and "Bar"), but can they still learn these mappings for other types of semantically-unrelated targets? To test this, we evaluate models in the SUL-ICL setup using varying semantically-unrelated targets in addition to Foo/Bar targets: numerical targets, alphabetical targets, and fruit targets.[10] For each target format, we also reverse the targets (e.g., $0 \rightarrow 1$ and $1 \rightarrow 0$) to verify that labels can be interchanged, at least within each set of labels. We experiment using natural language targets (i.e., regular ICL) for comparison.

Figure 14 shows model performance for each target type used.[11] We see that, in most cases, model performance stays relatively constant with respect to the target that is used. Additionally, there is no consistent difference between using natural language targets and using semantically-unrelated targets,

---

[10]While numerical targets such as "0" and "1" may have some semantic meaning in that "0" is often correlated with "negative" and "1" is often correlated with positive, our experiments show that this is not significant since reversing the 0/1 labels does not always hurt performance to the extent that the flipped-labels setting does.

[11]FP, ETHOS, and WSC contain fewer than 250 evaluation examples, so we use all available examples.

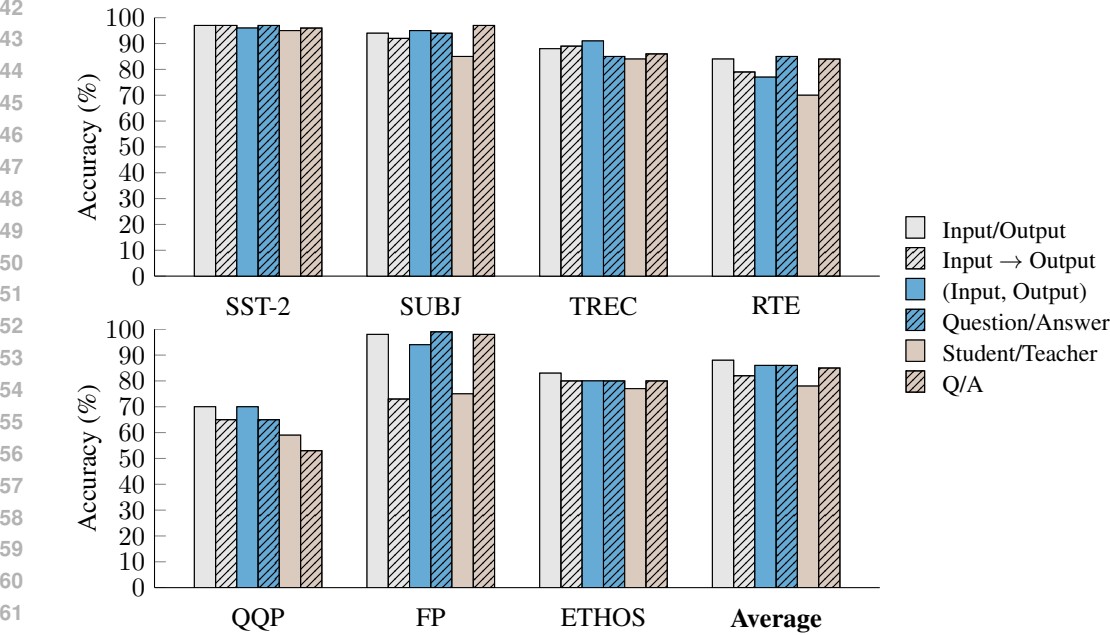

Figure 15: Model accuracy stays relatively consistent with respect to the input format used for SUL-ICL. Accuracy is calculated over 100 evaluation examples inputted to code-davinci-002 with $k = 16$ in-context exemplars per class.

which may suggest that given a large enough model and enough in-context exemplars, input–label mappings alone are enough to drive model performance. These findings demonstrate that for many types of semantically-unrelated targets, large models can still learn input–label mappings.

We can also see that some tasks are too difficult for the model to learn, regardless of whether natural language targets or SUL-ICL targets were used. Specifically, the model cannot significantly outperform random guessing on the QNLI and WSC datasets for any target type, and for this reason, we remove the QNLI and WSC datasets from other experiments.

### C.4 PROMPT TEMPLATES SHOWING INPUT–LABEL RELATIONSHIPS WORK

Can any prompt format be used for SUL-ICL as long as it clearly presents inputs and their respective labels? We explore this question by comparing the default Input/Output prompt template shown in Figure 10 with five additional formats, where [input] and [label] stand for the inputs and labels respectively (templates are shown in quotes).

- Input → Output: "[input]->[label]"
- (Input, Output): "[input], [label]"
- Question/Answer: "Question: [input] \n Answer: [label]"
- Student/Teacher: "Student: [input] \n Teacher: [label]"
- Q/A: "Q: [input] \n A: [label]"

In Figure 15, we show model performance for each of the input formats that we tested. We find that no input format is significantly better than any other input format, as the mean accuracy across all NLP tasks for all input formats (which ranges from 77.9% to 87.7%) is within $\pm 6.3\%$ of the mean (84.2%). These findings suggest that SUL-ICL may work across many simple formats that present input–label mappings, which may indicate that a factor to succeed in a SUL-ICL setup is that prompt templates should show a clear mapping between an input and its respective label.

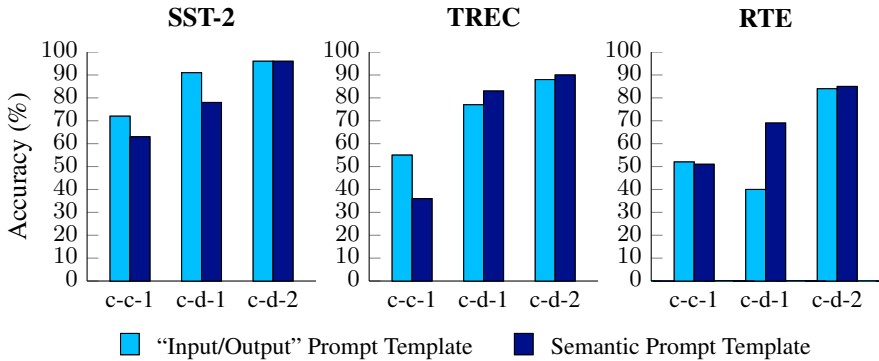

Figure 16: Small models do worse than large models do in the SUL-ICL setting when presented with semantically-relevant prompt templates. Accuracy is calculated over 100 evaluation examples inputted to Codex models with $k = 16$ in-context exemplars per class.

### C.5 Semantic prompt templates yield varying results depending on model size

In Appendix C.4, we did not test any prompt templates that include semantic information that is relevant to the task (e.g., using "Review: [input] \n Sentiment: [label]" for SST-2). We thus want to explore this setting in order to investigate whether models use semantic priors more or input–label mappings more they are given a semantically-relevant template.

We investigate this by using semantic prompt formats from Zhao et al. (2021) in the SUL-ICL setting and compare these results to the results from using our default "Input/Output" prompt template. We run these experiments on the SST-2, TREC, and RTE datasets—the datasets in our paper that intersect with those used in Zhao et al. (2021)—and we evaluate on the Codex model family.

As shown in Figure 16, we find that the smallest Codex model (code-cushman-001) sees performance drop across all tested datasets when switching to semantically-relevant prompt templates. The largest Codex model (code-davinci-002), on the other hand, is relatively unaffected by the change, while the middle Codex model (code-davinci-001) experiences performance changes that vary across datasets.

These results suggest that small models get worse at learning input–label mappings when presented with semantically-relevant prompts, perhaps because seeing semantically-charged words encourages the model to try to utilize semantic priors rather than learn input–label mappings in-context. We also see that large models may be more robust to these inputs—their performance being unaffected by the change indicates that despite seeing the semantic prompt templates, they are still able to learn the semantically-unrelated input–label mappings in-context.

### C.6 Large models are robust to out-of-distribution datasets

Tran et al. (2022) previously showed that model scale improves robustness to out-of-distribution (OOD) datasets where the input distribution of text for a given task changes. We aim to analyze whether this behavior is present in the SUL-ICL setting. In this experiment, we combine examples from SST-2 and the Rotten Tomatoes dataset (Pang & Lee, 2005, **RT**)—which is also a sentiment analysis dataset—and prompt the model with in-context exemplars from one dataset while evaluating it on examples from the other dataset. We then test InstructGPT models in a SUL-ICL environment using these varied input distributions.

As shown in Table 2, we see that small models (e.g., text-ada-001 and text-babbage-001) suffer from significant performance drops of up to $36\%$ when OOD datasets are used. Large models (e.g., text-curie-001 and text-davinci-001), on the other hand, do not suffer from these drops, with text-curie-001 only seeing a $4\%$ decrease in accuracy and text-davinci-001 seeing no significant change in accuracy. These results suggest that robustness to OOD datasets emerges with scale in the SUL-ICL setup, implying that this behavior could be related to the presentation of input–label mappings (something that both regular in-context learning and SUL-ICL share) and not necessarily the availability of semantic targets (which SUL-ICL lacks).

| Dataset | a-1 | b-1 | c-1 | d-1 |
|---|---|---|---|---|
| SST-2 Only (Baseline) | 80 | 91 | 94 | 93 |
| SST-2 (In-Context) + RT (Eval) | 54 | 63 | 90 | 93 |
| RT (In-Context) + SST-2 (Eval) | 44 | 61 | 90 | 92 |

Table 2: Robustness to out-of-distribution datasets in the SUL-ICL setup emerges with model scale. Accuracy is calculated over 100 evaluation examples with $k = 16$ in-context exemplars per class. "In-Context": examples used as in-context exemplars. "Eval": examples used as evaluation examples.

# D    FULL EXPERIMENTAL RESULTS

## D.1    THE FLIPPED LABELS SETTING

Here, we present per-dataset results for each model family after flipping labels for in-context exemplars, as described in Section 3. In Figure 17, we plot model accuracy with respect to the proportion of labels that we flip for each dataset and for each model family. We exclude the RTE dataset for LLM models because the prompts from this dataset at $k = 16$ in-context exemplars per class consistently exceed the maximum-allowable context length.

For many model families, we see that large models have better performance than small models do at 0% flipped labels, but that flipping more labels results in performance drops for large models but not for small models. This trend is especially true for the InstructGPT model family and, to a lesser extent, the Codex and LLM model families. The base GPT-3 model family, on the other hand, does not see this trend happen for most tasks, which is likely due to the fact that even the large models in this model family have trouble outperforming random guessing for many tasks. For example, the largest GPT-3 model (davinci) only achieves guessing accuracy on the QQP and RTE datasets, while the largest InstructGPT and Codex models both achieve $80\%+$ accuracy on these two tasks.

We find that many model families exhibit this behavior on the FP, RTE, and ETHOS datasets. Conversely, the SUBJ dataset seems to show that model performance drops across all model families and for all models within each model family, a result that suggests that it is easier for models to flip their predictions to follow flipped labels for this task, even if the model is small. It is unclear why this task in particular encourages flipping predictions to follow flipped labels more than other tasks do.

## D.2    THE SUL-ICL SETTING

In this section, we show per-dataset results for each model family after converting prompts to our SUL-ICL setup described in Section 4. Figure 18 gives a per-dataset overview of the performance differences between using SUL-ICL labels and using natural language labels as described in Section 4. We exclude the RTE dataset for LLM models because the prompts from this dataset at $k = 16$ in-context exemplars per class consistently exceed the maximum allowable context length. We find that for InstructGPT, Codex, and LLM models, large models see less of a performance drop than small models do when switching from natural language targets to semantically-unrelated targets, implying that they are more capable of learning input–label mappings when semantic priors are unavailable. Conversely, base GPT-3 models do not seem to follow the same trend, specifically in the case of davinci, which (on many tasks) sees the largest performance drops when using SUL-ICL targets despite being the largest model in the family. It is unclear why davinci seems to be the only large model that is not capable of learning input–label mappings in the SUL-ICL setup, though this behavior is consistent with davinci behaving similarly to small models as described in Section 3.

In Figure 19, we show per-dataset results for model accuracy with respect to the number of in-context exemplars provided. We do not run experiments on InstructGPT models and davinci in order to reduce cost. Lines do not always extend to $k = 32$ due to context-length constraints. These results indicate that for many datasets and model families, larger models are better at utilizing in-context exemplars in a SUL-ICL setup than small models are. This suggests that larger language models are more capable than small language models are at learning input–label mappings using the exemplars presented in-context rather than using prior knowledge from pretraining.

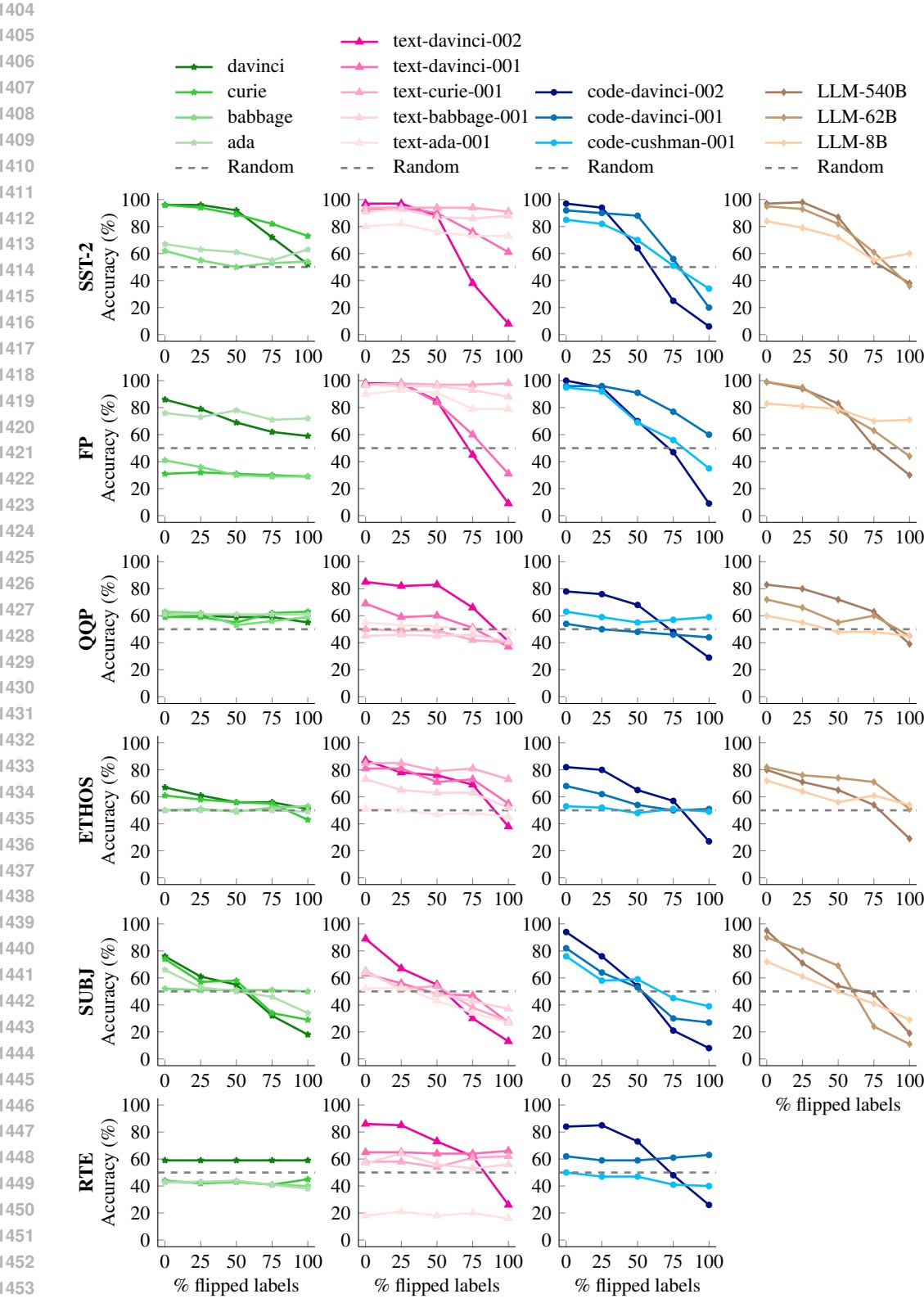

Figure 17: Larger models are better able to override semantic meanings when presented with flipped labels than smaller models are for many datasets and model families. Accuracy is calculated over 100 evaluations examples per dataset with $k = 16$ in-context exemplars per class.

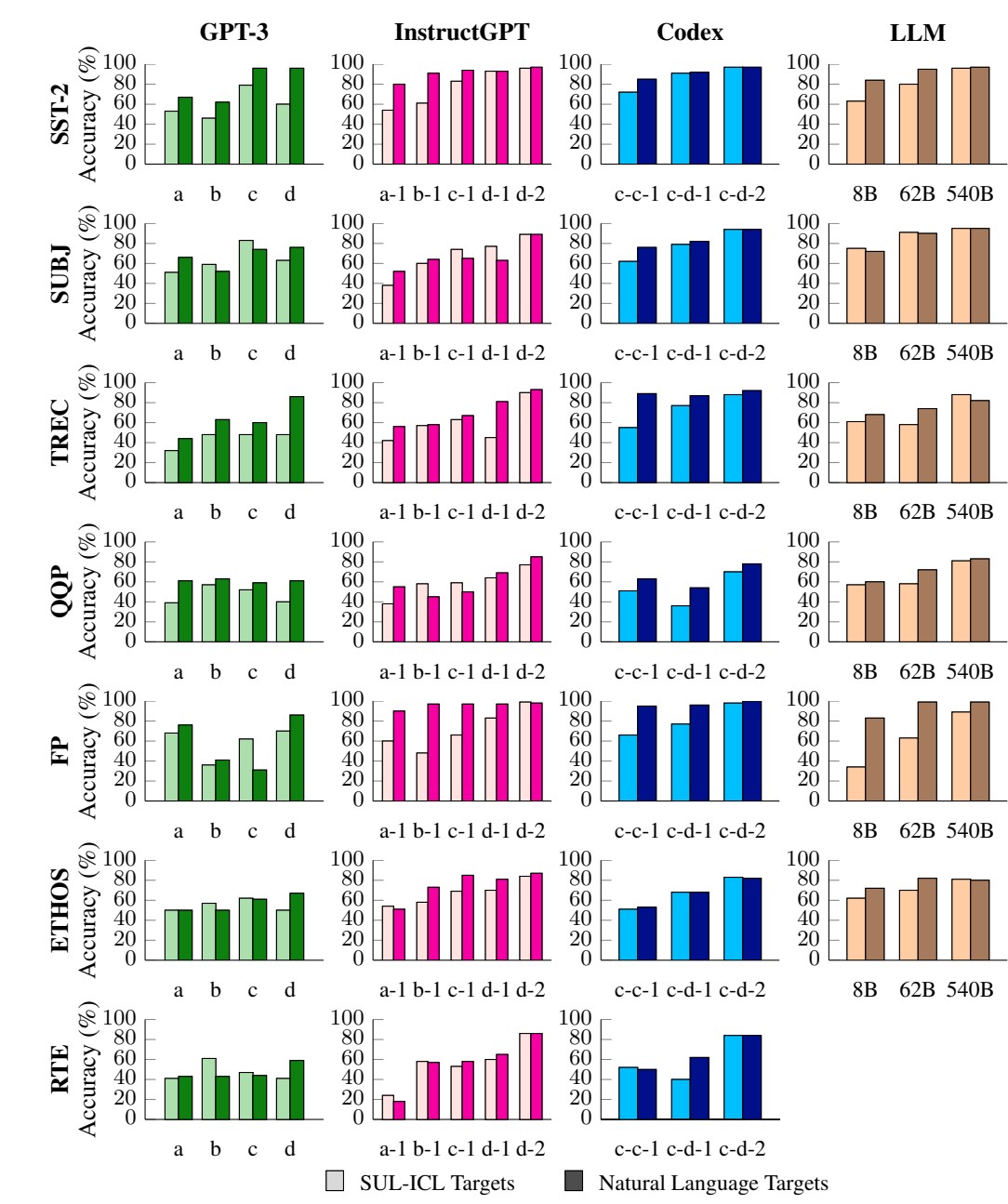

Figure 18: For many datasets and model families, performance decreases more for small models than it does for large models when using semantically-unrelated targets instead of natural language targets. Accuracy is calculated over 100 evaluation examples with $k = 16$ in-context exemplars per class.

## D.3 INSTRUCTION TUNING

We compare LLM and IT-LLM model behaviors on a per-dataset level as an extension of Section 5. First, we show model behavior in the SUL-ICL setting in Figure 20, finding that for the SST-2, QQP, RTE, and ETHOS datasets, IT-LLM models achieve higher performance than their respective LLM models. On the SST-2 dataset in particular, IT-LLM-8B outperforms LLM-8B by 28% and even outperforms LLM-62B by 2%. There are some datasets, however, for which instruction tuning seemed to decrease performance (e.g., LLM-8B outperforms IT-LLM-8B on SUBJ by 23%). These results indicate that for many tasks, instruction tuning increases the model's capacity to learn input–label

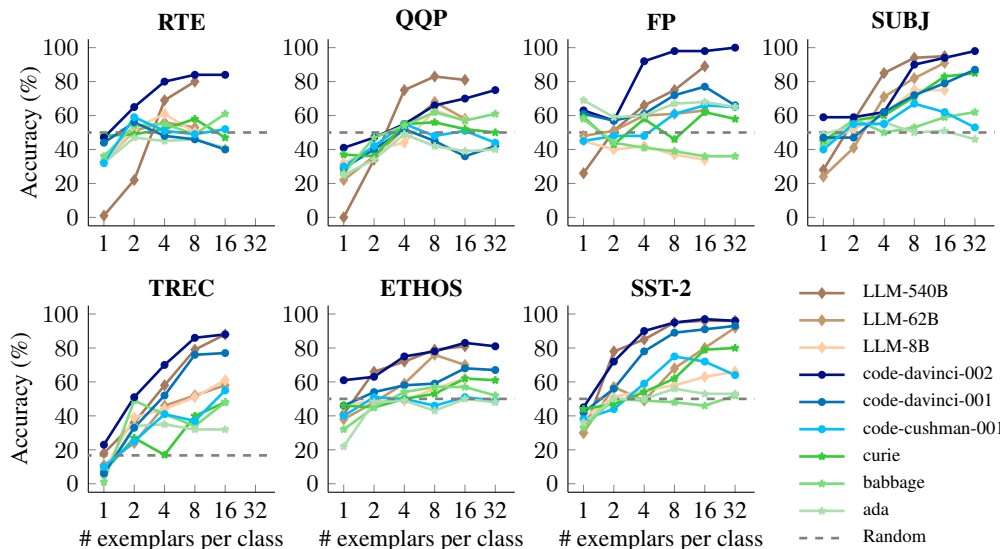

Figure 19: For many datasets and model families, large language models are better at using in-context exemplars to learn input–label mappings than small language models are. Accuracy is calculated over 100 examples in the SUL-ICL setup.

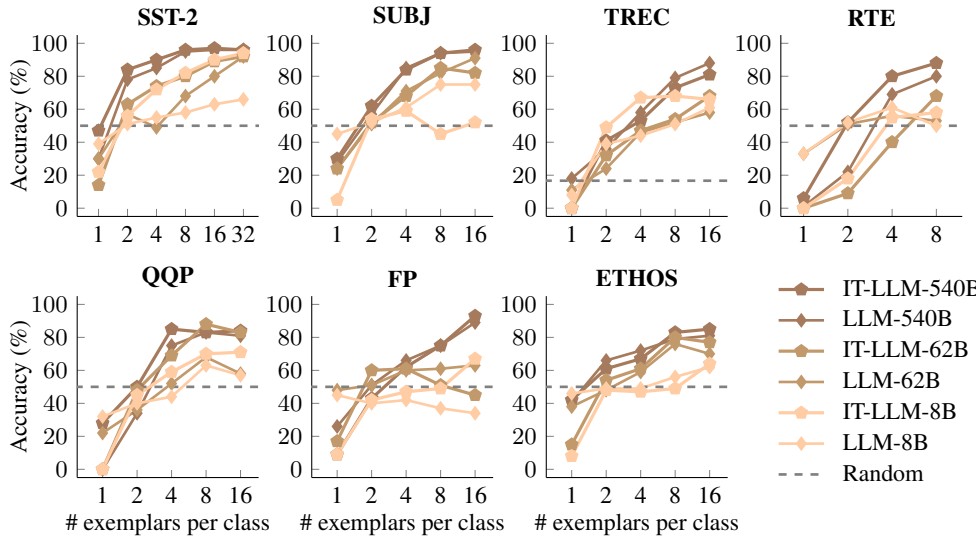

Figure 20: For many datasets, instruction-tuned language models are better at learning input–label mappings than pretraining-only language models are. Accuracy is calculated over 100 evaluation examples in the SUL-ICL setup.

mappings in-context (though there are some exceptions), which follows the findings from Section 5. We also found that across most datasets, IT-LLM does worse than LLM and scores close to 0% accuracy when given one in-context exemplar per class, yet this does not seem to be the case when two or more in-context exemplars per class are presented. Why this occurs is unknown, but it may indicate that IT-LLM does not give a response that is part of the target set of responses (e.g., does not output "Foo" or "Bar") in a 1-shot SUL-ICL setting.

In Figure 21, we show results for LLM and IT-LLM in the flipped-label setting. For all datasets,[12] we find that every IT-LLM model achieves better performance than its respective LLM model. LLM

---

[12]We do not run this experiment for the RTE dataset because prompts consistently exceed the context length.

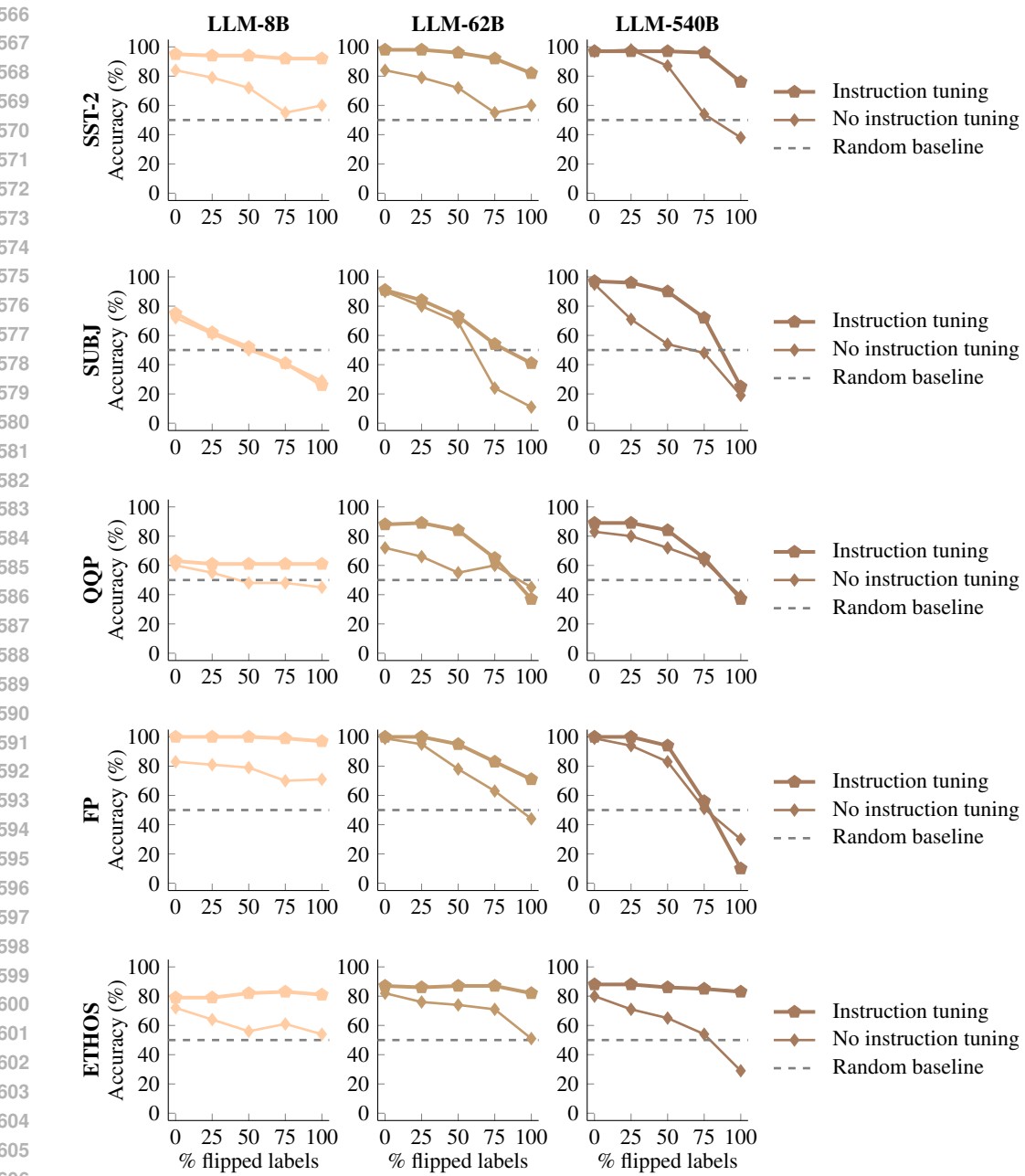

Figure 21: For all datasets and model sizes, instruction-tuned language models are worse than pretraining-only language models are at learning to override their semantic priors when presented with flipped labels in-context. Accuracy is calculated over 100 evaluation examples with $k = 16$ in-context exemplars per class and averaged across all datasets.

models notably have lower accuracy when more labels are flipped, which means that LLM models are better than IT-LLM models are at learning flipped input–label mappings presented in context, suggesting that it is harder for IT-LLM models to override semantic priors. This suggests that instruction tuning reinforces the model's semantic priors or gives it more semantic priors, making it more difficult for the model to override its prior knowledge.

---

**Algorithm 1** Generating one evaluation example for $N$-dimensional linear classification ($y = a_1 x_1 + ... + a_N x_N$) with $k$ in-context exemplars per class. Random $N$-D vectors are generated using `np.random.randint()`.

---

1: **procedure** GENERATEEVAL($N, k$)
2:     $a \leftarrow$ random $N$-D vector                                        ▷ Ground-truth coefficients
3:     $p \leftarrow$ random $N$-D vector                                         ▷ A pivot point
4:     $t = \langle a, p \rangle$                               ▷ Threshold between positive and negative examples
5:     $x_{train} \leftarrow [\,]$, $y_{train} \leftarrow [\,]$
6:     **for** $i \leftarrow 1$ to $k$ **do**                                    ▷ $2k$ in-context exemplars
7:         $x_+ \leftarrow$ random $N$-D vector conditioned on $\langle x_+, a \rangle > t$     ▷ Positive example
8:         $x_- \leftarrow$ random $N$-D vector conditioned on $\langle x_-, a \rangle \leq t$     ▷ Negative example
9:         $x_{train} \leftarrow x_{train} + [x_+, x_-]$
10:        $y_{train} \leftarrow y_{train} + [1, -1]$
11:    **end for**
12:    $x_{eval} \leftarrow$ random $N$-D vector
13:    $y_{eval} \leftarrow 1$ **if** $\langle x_{eval}, a \rangle > t$, **else** $-1$
14:    **return** $x_{train}, y_{train}, x_{eval}, y_{eval}$
15: **end procedure**

---

## E  FULL PROMPT EXAMPLES

In Appendix E.1–Appendix E.7, we include an example of a full few-shot prompt for each of the seven datasets used in the main paper. We show prompts with $k = 16$ in-context exemplars per class and the Input/Output prompt template from Appendix C.4 (our default experimental setup) and natural language targets (i.e., regular ICL). Prompts in a SUL-ICL and flipped-label ICL setup can be obtained by swapping labels with the desired labels (e.g., replacing "Negative Sentiment" with "Foo" and "Positive Sentiment" with "Bar" to convert SST-2 in a regular ICL setup to SST-2 in a SUL-ICL setup). Prompts (especially from the ETHOS dataset) may contain offensive language—note that all examples are directly taken from the existing datasets as referenced in Appendix B.

In Appendix E.8, we provide an example of a full prompt for the linear classification task from Section 6. This prompt uses the same default experimental setup as the prompts from Appendix E.1–Appendix E.7 but uses SUL-ICL targets since we only used this dataset in SUL-ICL settings. For reference, negative examples are labeled "Foo" and positive examples are labeled "Bar" (see Algorithm 1 for details about negative and positive examples).

### E.1  SST-2

**Prompt:**

Input: a pale imitation

Output: Negative Sentiment

Input: carries you along in a torrent of emotion

Output: Positive Sentiment

Input: trashy time

Output: Negative Sentiment

Input: all the complexity and realistic human behavior of an episode of general hospital

Output: Negative Sentiment

Input: hold dear about cinema ,

Output: Positive Sentiment

Input: inauthentic

Output: Negative Sentiment

---

Input: feels like very light errol morris , focusing on eccentricity but failing , ultimately , to make something bigger out of its scrapbook of oddballs

Output: Negative Sentiment

Input: with purpose and finesse

Output: Positive Sentiment

Input: feel a nagging sense of deja vu

Output: Positive Sentiment

Input: and mawkish dialogue

Output: Negative Sentiment

Input: , but i believe a movie can be mindless without being the peak of all things insipid .

Output: Negative Sentiment

Input: it does elect to head off in its own direction

Output: Positive Sentiment

Input: falls flat as a spoof .

Output: Negative Sentiment

Input: charm , cultivation and devotion

Output: Positive Sentiment

Input: it has some special qualities and the soulful gravity of crudup 's anchoring performance .

Output: Positive Sentiment

Input: the work of a genuine and singular artist

Output: Positive Sentiment

Input: bravado – to take an entirely stale concept and push it through the audience 's meat grinder one more time

Output: Negative Sentiment

Input: and unfunny tricks

Output: Negative Sentiment

Input: that made mamet 's " house of games " and last fall 's " heist " so much fun

Output: Positive Sentiment

Input: is a light , fun cheese puff of a movie

Output: Positive Sentiment

Input: a generic family comedy unlikely to be appreciated by anyone outside the under-10 set .

Output: Negative Sentiment

Input: , treasure planet is truly gorgeous to behold .

Output: Positive Sentiment

Input: the bai brothers have taken an small slice of history and opened it up for all of us to understand , and they 've told a nice little story in the process

Output: Positive Sentiment

Input: sentimental cliches

Output: Negative Sentiment

Input: the demented mind

Output: Negative Sentiment

Input: most certainly has a new career ahead of him

Output: Positive Sentiment

Input: while this film has an ' a ' list cast and some strong supporting players , the tale – like its central figure , vivi – is just a little bit hard to love .

Output: Negative Sentiment

Input: an exhausted , desiccated talent

Output: Negative Sentiment

Input: a relentless , bombastic and ultimately empty world war ii action

Output: Negative Sentiment

Input: the sheer joy and pride

Output: Positive Sentiment

Input: so larger than life

Output: Positive Sentiment

Input: to its superior cast

Output: Positive Sentiment

Input: one of the more intelligent children 's movies to hit theaters this year .

Output:

**Answer:**

Positive Sentiment

E.2   SUBJ

**Prompt:**

Input: an impossible romance , but we root for the patronized iranian lad .

Output: Subjective Sentence

Input: . . . plays like a badly edited , 91-minute trailer ( and ) the director ca n't seem to get a coherent rhythm going . in fact , it does n't even seem like she tried .

Output: Subjective Sentence

Input: the stunt work is top-notch ; the dialogue and drama often food-spittingly funny .

Output: Subjective Sentence

Input: no such thing may be far from perfect , but those small , odd hartley touches help you warm to it .

Output: Subjective Sentence

Input: a positively thrilling combination of ethnography and all the intrigue , betrayal , deceit and murder of a shakespearean tragedy or a juicy soap opera .

Output: Subjective Sentence

Input: it trusts the story it sets out to tell .

Output: Subjective Sentence

Input: so , shaun goes to great lengths with a little help from his girlfriend ashley and his drugged-out loser brother lance to get into stanford any way they see fit .

Output: Objective Sentence

Input: are they illusions , visions from the past , ghosts - or is it reality ?

Output: Objective Sentence

Input: all the amped-up tony hawk-style stunts and thrashing rap-metal ca n't disguise the fact that , really , we 've been here , done that .

Output: Subjective Sentence

Input: a master at being everybody but himself he reveals to his friend and confidant saiid ( isa totah ) the truth behind his struggles .

Output: Objective Sentence

Input: three families , living in a three storey building , leave for their summer vacations .

Output: Objective Sentence

Input: the directing and story are disjointed , flaws that have to be laid squarely on taylor 's doorstep . but the actors make this worth a peek .

Output: Subjective Sentence

Input: together , they team up on an adventure that would take them to some very unexpected places and people .

Output: Objective Sentence

Input: jacquot 's rendering of puccini 's tale of devotion and double-cross is more than just a filmed opera . in his first stab at the form , jacquot takes a slightly anarchic approach that works only sporadically .

Output: Subjective Sentence

Input: evil czar and his no-less-evil sidekick general with the help of the local witch yaga try to eliminate fedot by giving him more and more complex quests and to take marusya to tsar 's palace .

Output: Objective Sentence

Input: the clues are few and time is running out for the students of rogers high school .

Output: Objective Sentence

Input: seducing ben is only beginning ; she becomes his biggest " fan " and most unexpected nightmare , as her obsessions quickly spiral out of control into betrayal , madness and , ultimately , murder .

Output: Objective Sentence

Input: but despite his looks of francis , he indeed is henry ( timothy bottoms ) , a man with a much better character than patricia ever could have dreamt of .

Output: Objective Sentence

Input: the actors pull out all the stops in nearly every scene , but to diminishing effect . the characters never change .

Output: Subjective Sentence

Input: in 1946 , tests began using nazi v-1 " buzz bombs " launched from the decks of american diesel submarines .

Output: Objective Sentence

Input: a clichM-id and shallow cautionary tale about the hard-partying lives of gay men .

Output: Subjective Sentence

Input: the characters search for meaning in capricious , even dangerous sexual urges . the irony is that the only selfless expression of love may be the failure to consummate it .

Output: Subjective Sentence

Input: meanwhile , chris 's radio horoscopes seem oddly personal , and the street musicians outside uwe 's restaurant keep getting more numerous .

Output: Objective Sentence

Input: battling his own demons he realizes he is just like the rest of us : good and evil .

Output: Objective Sentence

Input: or so he tells bobby ( alex feldman ) the eighteen year old male hustler smith employs for company .

Output: Objective Sentence

Input: two brothers along with an ensemble of fresh talent made all this possible and were brought into the light .

Output: Objective Sentence

Input: sandra bullock and hugh grant make a great team , but this predictable romantic comedy should get a pink slip .

Output: Subjective Sentence

Input: nora is not interested in foreign political smalltalk , she is after government secrets .

Output: Objective Sentence

Input: godard has never made a more sheerly beautiful film than this unexpectedly moving meditation on love , history , memory , resistance and artistic transcendence .

Output: Subjective Sentence

Input: elmo touts his drug as being 51 times stronger than coke . if you 're looking for a tale of brits behaving badly , watch snatch again . it 's 51 times better than this .

Output: Subjective Sentence

Input: culled from nearly two years of filming , the documentary 's candid interviews , lyric moments of grim beauty , and powerful verite footage takes us beyond the usual stereotypes of the rap world and into the life of tislam milliner , a struggling rapper who 's ambitious to make it out of the " hood " .

Output: Objective Sentence

Input: i wish windtalkers had had more faith in the dramatic potential of this true story . this would have been better than the fiction it has concocted , and there still could have been room for the war scenes .

Output: Subjective Sentence

Input: has lost some of the dramatic conviction that underlies the best of comedies . . .

**Answer:**

Subjective Sentence

## E.3   TREC

**Prompt:**

Input: What is the real name of the singer , Madonna ?

Output: Human Being

Input: What snack food has ridges ?

Output: Entity

Input: How do you correctly say the word ' qigong ' ?

Output: Description and Abstract Concept

Input: Which Bloom County resident wreaks havoc with a computer ?

Output: Human Being

Input: What does HIV stand for ?

Output: Abbreviation

Input: What does Warner Bros. call a flightless cuckoo ?

Output: Entity

Input: What causes pneumonia ?

Output: Description and Abstract Concept

Input: What were hairy bank notes in the fur trade ?

Output: Entity

Input: Where is the world 's most active volcano located ?

Output: Location

Input: What is the origin of the word trigonometry ?

Output: Description and Abstract Concept

Input: What is the city in which Maurizio Pellegrin lives called ?

Output: Location

Input: What is in baby powder and baby lotion that makes it smell the way it does ?

Output: Description and Abstract Concept

Input: What actress 's autobiography is titled Shelley : Also Known as Shirley ?

Output: Human Being

Input: What does the E stand for in the equation E=mc2 ?

Output: Abbreviation

Input: What Southern California town is named after a character made famous by Edgar Rice Burroughs ?

Output: Location

Input: What is the student population at the University of Massachusetts in Amherst ?

Output: Numeric Value

Input: Where did makeup originate ?

Output: Location

Input: What did Englishman John Hawkins begin selling to New World colonists in 1562 ?

Output: Entity

Input: Who did Napolean defeat at Jena and Auerstadt ?

Output: Human Being

Input: What country 's royal house is Bourbon-Parma ?

Output: Location

Input: Where is the Thomas Edison Museum ?

Output: Location

Input: What group asked the musical question Do You Believe in Magic ?

Output: Human Being

Input: When are sheep shorn ?

Output: Numeric Value

Input: How many propellers helped power the plane the Wright brothers flew into history ?

Output: Numeric Value

Input: When was Queen Victoria born ?

Output: Numeric Value

Input: What does the word LASER mean ?

Output: Abbreviation

Input: On which dates does the running of the bulls occur in Pamplona , Spain ?

Output: Numeric Value

Input: McCarren Airport is located in what city ?

Output: Location

Input: What does VCR stand for ?

Output: Abbreviation

Input: What does RCA stand for ?

Output: Abbreviation

Input: What J.R.R. Tolkien book features Bilbo Baggins as the central character ?

Output: Entity

Input: What is the abbreviated form of the National Bureau of Investigation ?

Output: Abbreviation

Input: Who painted " Soft Self-Portrait with Grilled Bacon " ?

Output: Human Being

Input: Where is the Virtual Desk Reference ?

Output: Location

Input: Where is Trinidad ?

Output: Location

Input: Why is Indiglo called Indiglo ?

Output: Description and Abstract Concept

Input: What Asian leader was known as The Little Brown Saint ?

Output: Human Being

Input: What do I need to learn to design web pages ?

Output: Description and Abstract Concept

Input: What U.S. city was named for St. Francis of Assisi ?

Output: Location

Input: What shape-shifting menace did Rom come to Earth to fight ?

Output: Entity

Input: What does Ms. , Miss , and Mrs. stand for ?

Output: Abbreviation

Input: What is the abbreviation of the company name ' General Motors ' ?

Output: Abbreviation

Input: What was the name of the orca that died of a fungal infection ?

Output: Entity

Input: When did the Carolingian period begin ?

Output: Numeric Value

Input: What architect originated the glass house designed the Chicago Federal Center had a philosophy of " less is more , " and produced plans that were the forerunner of the California ranch house ?

Output: Human Being

Input: How high must a mountain be to be called a mountain ?

Output: Numeric Value

Input: What does snafu stand for ?

Output: Abbreviation

Input: Who shared a New York City apartment with Roger Maris the year he hit 61 home runs ?

Output: Human Being

Input: What is the location of McCarren Airport ?

Output: Location

Input: How many people die of tuberculosis yearly ?

Output: Numeric Value

Input: What is IOC an abbreviation of ?

Output: Abbreviation

Input: What is HTML ?

Output: Abbreviation

Input: What does the " blue ribbon " stand for ?

Output: Abbreviation

Input: What does the term glory hole mean ?

Output: Description and Abstract Concept

Input: What does the abbreviation cwt. ?

Output: Abbreviation

Input: How many students attend the University of Massachusetts ?

Output: Numeric Value

Input: Who was the captain of the tanker , Exxon Valdez , involved in the oil spill in Prince William Sound , Alaska , 1989 ?

Output: Human Being

Input: What should the oven be set at for baking Peachy Oat Muffins ?

Output: Entity

Input: What bread company used to feature stickers of the Cisco Kid on the ends of their packages ?

Output: Human Being

Input: Why do airliners crash vs. gliding down ?

Output: Description and Abstract Concept

Input: What is a fear of fish ?

Output: Entity

Input: Which country did Hitler rule ?

Output: Location

Input: What does A&W of root beer fame stand for ?

Output: Abbreviation

Input: How does a hydroelectric dam work ?

Output: Description and Abstract Concept

Input: What year did the Vietnam War end ?

Output: Numeric Value

Input: What are some children 's rights ?

Output: Description and Abstract Concept

Input: What is Colin Powell best known for ?

Output: Description and Abstract Concept

Input: What is the largest island in the Mediterranean Sea ?

Output: Location

Input: What is a fear of weakness ?

Output: Entity

Input: What 's the world 's most common compound ?

Output: Entity

Input: Why do people in the upper peninsula of Michagin say " eh ? " ?

Output: Description and Abstract Concept

Input: Why do many Native American students not complete college ?

Output: Description and Abstract Concept

Input: When are the Oscars Academy Awards in 1999 ?

Output: Numeric Value

Input: Where can I get cotton textiles importer details ?

Output: Location

Input: What is a fear of childbirth ?

Output: Entity

Input: When were camcorders introduced in Malaysia ?

Output: Numeric Value

Input: How long does a fly live ?

Output: Numeric Value

Input: What is the largest office block in the world ?

Output: Location

Input: How long does the average domesticated ferret live ?

Output: Numeric Value

Input: Which magazine is " fine entertainment for men " ?

Output: Entity

Input: What does JESSICA mean ?

Output: Abbreviation

Input: Who invented the vacuum cleaner ?

Output: Human Being

Input: When is the Sun closest to the Earth ?

Output: Numeric Value

Input: What is the abbreviation of the International Olympic Committee ?

Output: Abbreviation

Input: What 's the name of the tiger that advertises for Frosted Flakes cereal ?

Output: Entity

Input: What Caribbean island is northeast of Trinidad ?

Output: Location

Input: What deck of cards includes the Wheel of Fortune , the Lovers , and Death ?

Output: Entity

Input: Who played for the Chicago Bears , Houston Oilers and Oakland Raiders in a 26-year pro football career ?

Output: Human Being

Input: How many varieties of twins are there ?

Output: Numeric Value

Input: What " marvelous " major-league baseball player is now a spokesman for a beer company ?

Output: Human Being

Input: What was the claim to fame of Explorer I , launched February 1 , 1958 ?

Output: Description and Abstract Concept

Input: What do the number 1 , 2 , and 4 mean on Dr. Pepper bottles ?

Output: Description and Abstract Concept

Input: Who is Edmund Kemper ?

Output: Human Being

Input: What are differences between 1980 and 1990 ?

Output: Description and Abstract Concept

Input: What 2 statues did France give to other countries ?

Output: Entity

Input: Whose biography by Maurice Zolotow is titled Shooting Star ?

Output: Human Being

Input: What kind of gas is in a fluorescent bulb ?

Output:

**Answer:**

Entity

E.4    QQP

**Prompt:**

Input: Why did Indian Government introduced 2000 note instead of the new 1000 note? Meanwhile, they introduced the new 500 note for old 500 note.

If 500 and 1000 notes are banned then why are new 500 and 2000 notes being introduced?

Output: Duplicate

Input: Where can I get a free iTunes gift card without doing a survey or download?

How can I download the Itunes gift card generator with no surveys?

Output: Not a duplicate

Input: Is petroleum engineering still a good major?

Is the petroleum engineering major still worthy to choose today? And how about in the future 2020-2025?

Output: Duplicate

Input: Is Minecraft Turing complete?

Why is Minecraft so popular?

Output: Not a duplicate

Input: What are some HR jobs in Mumbai?

How do I get a HR job in Bangalore?

Output: Not a duplicate

Input: To which caste and category does the surname Saini belong to?

"Which caste (General/OBC/SC/ST) does ""Bera"" surname belongs to?"

Output: Not a duplicate

Input: Who are burning the schools in Kashmir and why?

Why are separatists burning schools in Kashmir?

Output: Duplicate

Input: How do I remove onclick ads from Chrome?

How do I reduce the CPA on my Facebook Ads?

Output: Not a duplicate

Input: How should I start learning Python?

How can I learn advanced Python?

Output: Duplicate

Input: How do I stop feeling sad?

How do I stop feeling sad about nothing?

Output: Not a duplicate

Input: How can you lose 10 pounds in 40 days?

What are some great diet plans to lose 10 pounds in 40 days?

Output: Duplicate

Input: What are job opportunities after completing one year of a HAL graduate apprenticeship?

What are some opportunities after completing one year of a HAL graduate apprenticeship?

Output: Duplicate

Input: Why did liquidprice.com fail?

Why did ArchiveBay.com fail?

Output: Not a duplicate

Input: Why is everyone on Quora obsessed with IQ?

Why are people on Quora so obsessed with people's high IQs?

Output: Duplicate

Input: I want to learn Chinese, which app is better for it?

I am basically Non IT Background.. I want learn course...Some of my friends suggested Linux and PLSql.. I want to know which is best option for me?

Output: Not a duplicate

Input: How is black money gonna go off with no longer the use of same 500 and 1000 notes?

How is discontinuing 500 and 1000 rupee note going to put a hold on black money in India?

Output: Duplicate

Input: How did Jawaharlal Nehru die? Was it really a sexually transmittable disease?

How can I become a great person like Jawaharlal Nehru?

Output: Not a duplicate

Input: What are the career option after completing of B.tech?

What are the career options available after completing a B.Tech?

Output: Duplicate

Input: What would be next strike from PM Modi after Demonetisation?

What will be the next move by PM Modi to improve India?

Output: Duplicate

Input: What should I do to beat loneliness?

How can I beat loneliness?

Output: Duplicate

Input: Dreams and Dreaming: What is your idea of Utopia?

Do you have any idea about lucid dreaming?

Output: Not a duplicate

Input: My boyfriend dumped me because I am not like other girls who wear makeup and fashionable clothes. What should I do?

How often do people stop wearing clothes because of wear, as opposed to them no longer being fashionable or other reasons?

Output: Not a duplicate

Input: Why does a persons taste change

What does caviar taste like?

Output: Not a duplicate

Input: Why is Sachin Tendulkar called a legend of cricket?

Why is Sachin Tendulkar still a legend of cricket?

Output: Duplicate

Input: What are some interesting examples on the availability heuristic?

What is heuristic search in AI?

Output: Not a duplicate

Input: How can I commit suicide without any pain?

What is best way to commit suicide painlessly?

Output: Duplicate

Input: How do I get started as a freelance web developer?

How can I best get started freelancing as a web developer and/or telecommute as a web developer?

Output: Not a duplicate

Input: What are some mind blowing gadgets for photography that most people don't know about?

What are some mind-blowing inventions gadgets that most people don't know about?

Output: Not a duplicate

Input: How can I lose weight safely?

What can I do to lose 20 pounds?

Output: Duplicate

Input: If Hitler's Germany hadn't attacked the Soviet Union, would the Allies have won WW2?

What would have happened if Germany had not attacked the Soviet Union in Operation Barbaross?

Output: Duplicate

Input: Is there any sort of root that I can use on my LG Phoenix 2?

How in the hell do I get this Android 6.0 LG Phoenix 2 (LG-k371) root access?

Output: Duplicate

Input: What is the price of booking Hardwell?

How does Hardwell on air make money?

Output: Not a duplicate

Input: Is theft at the threat of kidnapping and death acceptable? What if that money went to education and medicine for those who couldn't afford it?

If you were a cashier, and a young child wanted to buy an item for their terminally ill parent, and they couldn't quite afford it, would you give them the money?

Output:

**Answer:**

Not a duplicate

## E.5 FP

**Prompt:**

Input: Stora Enso Oyj said its second-quarter result would fall by half compared with the same period in 2007 .

Output: Negative

Input: Konecranes Oyj KCR1V FH fell 5.5 percent to 20.51 euros , the biggest fall since June .

Output: Negative

Input: Net sales of Finnish Sanoma Learning & Literature , of Finnish media group Sanoma , decreased by 3.6 % in January-June 2009 totalling EUR 162.8 mn , down from EUR 168.8 mn in the corresponding period in 2008 .

Output: Negative

Input: Finnish silicon wafers manufacturer Okmetic Oyj said it swung to a net profit of 4.9 mln euro $ 6.3 mln in the first nine months of 2006 from a net loss of 1.8 mln euro $ 2.3 mln a year earlier .

Output: Positive

Input: I am extremely delighted with this project and the continuation of cooperation with Viking Line .

Output: Positive

Input: Cash flow from operations rose to EUR 52.7 mn from EUR 15.6 mn in 2007 .

Output: Positive

Input: EPS for the quarter came in at 0.36 eur , up from 0.33 eur a year ago and ahead of forecast of 0.33 eur .

Output: Positive

Input: EBIT excluding non-recurring items , totalled EUR 67.8 mn , up from EUR 38.1 mn .

Output: Positive

Input: Profit for the period increased from EUR 2.9 mn to EUR 10.5 mn .

Output: Positive

Input: Net profit fell by almost half to +é 5.5 million from +é 9.4 million at the end of 2007 .

Output: Negative

Input: 17 March 2011 - Goldman Sachs estimates that there are negative prospects for the Norwegian mobile operations of Norway 's Telenor ASA OSL : TEL and Sweden 's TeliaSonera AB STO : TLSN in the short term .

Output: Negative

Input: Both operating profit and net sales for the three-month period increased , respectively from EUR15 .1 m and EUR131 .5 m , as compared to the corresponding period in 2005 .

Output: Positive

Input: Operating profit fell to EUR 20.3 mn from EUR 74.2 mn in the second quarter of 2008 .

Output: Negative

Input: Operating profit decreased to nearly EUR 1.7 mn , however .

Output: Negative

Input: Operating profit in the fourth quarter fell to EUR33m from EUR39m a year earlier .

Output: Negative

Input: Prices and delivery volumes of broadband products decreased significantly in 2005 .

Output: Negative

Input: The steelmaker said that the drop in profit was explained by the continuing economic uncertainty , mixed with the current drought in bank lending , resulting in a decline in demand for its products as customers find it increasingly difficult to fund operations .

Output: Negative

Input: The company 's scheduled traffic , measured in revenue passenger kilometres RPK , grew by just over 2 % and nearly 3 % more passengers were carried on scheduled flights than in February 2009 .

Output: Positive

Input: Diluted EPS rose to EUR3 .68 from EUR0 .50 .

Output: Positive

Input: LONDON MarketWatch – Share prices ended lower in London Monday as a rebound in bank stocks failed to offset broader weakness for the FTSE 100 .

Output: Negative

Input: The transactions would increase earnings per share in the first quarter by some EUR0 .28 .

Output: Positive

Input: The brokerage said 2006 has seen a ' true turning point ' in European steel base prices , with better pricing seen carrying through the second quarter of 2006 .

Output: Positive

Input: However , the orders received during the period under review fell by 17 % quarter-on-quarter from the EUR 213 million recorded in the second quarter of 2010 .

Output: Negative

Input: Operating profit totalled EUR 9.0 mn , down from EUR 9.7 mn in the first half of 2008 .

Output: Negative

Input: Finnish Bank of +àland reports operating profit of EUR 2.2 mn in the first quarter of 2010 , down from EUR 6.3 mn in the corresponding period in 2009 .

Output: Negative

Input: Profit after taxes was EUR 0.1 mn , compared to EUR -0.4 mn the previous year .

Output: Positive

Input: Finnish IT consultancy Satama Interactive Oyj posted a net profit of 1.4 mln euro $ 2.0 mln for the first nine months of 2007 , compared to a net loss of 462,000 euro $ 664,000 for the same period of 2006 .

Output: Positive

Input: Shares in Royal and Sun Alliance continued to slide back from a 12-month high of 172p last month , after a potential suitor ruled itself out of a takeover bid .

Output: Negative

Input: Olvi , which controls a 94 percent stake in Ragutis through A. Le Coq , said in its annual report published earlier this year that the Lithuanian brewery 's sales reached 15.04 million euros last year , a rise of 20.4 percent from 12.49 million euros in 2004 .

Output: Positive

Input: Elisa said mobile subscriptions grew 7 percent during 2007 , mainly because of customers adopting so-called third generation mobile technology .

Output: Positive

Input: For the first nine months of 2010 , the company 's net profit rose to EUR41m from EUR30m for the corresponding period of 2009 .

Output: Positive

Input: Operating profit for 2009 lower than outlook published earlier .

Output: Negative

Input: The company slipped to an operating loss of EUR 2.6 million from a profit of EUR 1.3 million .

Output:

**Answer:**

Negative

### E.6    ETHOS

**Prompt:**

Input: i like dead refugees on my shore

Output: Hate Speech

Input: Fuck these generate vodka niggers. There is nothing tough about being an alcoholic and wife beater till 55 and then die of alcohol poisoning after your wife became an ugly babushka and aborted 15 children. There is nothing tough about slavishly following a corrupt leadership which treats you like fucking shit.

Output: Hate Speech

Input: Support from China , we also fight the cancer religion in Xinjiang , despite all the differences between our two races, we have one unite enemy that is Islam.

Output: Hate Speech

Input: There was a protest today in Birmingham against hate. It was so great to see Muslims and Gays marching together in unity.

Output: Hate Speech

Input: Very sad for those people. i was a young boy in the 90s when Ceausescu fell in Romania and the inflation skyrocketed. But here they are on another level and looks like there is no end soon. How can this happen to one of the richest country in the world? Venezuela has the largest resource of oil in the world. When you look at Qatar, UAE, Saudi Arabia they done miracles. It looks like is not enough to have oil, you have to know what to do with it.

Output: Not Hate Speech

Input: guy is low IQ af

Output: Not Hate Speech

Input: Here in 2019. May history be on the right side this time.

Output: Not Hate Speech

Input: When I see men running with those plastic legs I laugh my ass out so hard I fart some times

Output: Hate Speech

Input: The rotten, nasty bastards!

Output: Not Hate Speech

Input: or when the man walked up to the little girl with the gay pride flag i wanted to jump into the video and punch him but ya know thats assault

Output: Hate Speech

Input: Fucking idiots. Keep them out. They change the culture. Take over small towns. Vote in their own people to rule the town. And start enforcing their religious beliefs on everyone. It CANNOT FUCKING HAPPEN. Stop them now while you still can.

Output: Hate Speech

Input: OMG Lucas !! Such a great kid, such a big grow

Output: Not Hate Speech

Input: Im going to choke you with your cross necklace idiotic religious pig

Output: Hate Speech

Input: I very much appreciate the Asian woman's smarts.

Output: Not Hate Speech

Input: fuck bulgaria stupid country

Output: Hate Speech

Input: well it's clear now why europeans can't differ niggers and shitskins from human

Output: Hate Speech

Input: I notice that the LGBT cancer is starting to slowly affect SA countries like Japan.

Output: Hate Speech

Input: You guys I fell so bad, I hope this don't happen again. Im a Islam

Output: Not Hate Speech

Input: My smile is beaming, my skin is gleaming! YUH

Output: Not Hate Speech

Input: All theses people talking about Nigeria, since when have they cared about nigeria? They find it now to care about Nigeria to be relevenat, those who were killed were farmers and Nigeria is living peacefully with it's Muslims. Sure there are terrorist attacks every now and then, but we must remember that those who do kill the innocent are not Muslims. Many terrorists or groups use religion as an excuse. How come Isis are killing their own people? How come Iraqi forces are trying to drive isis Muslims out? Why? Cause they are scum who are not Muslims.

Output: Not Hate Speech

Input: The Whites Have dislike the documentary because they are mad at the truth. #WhiteDevils

Output: Hate Speech

Input: Fuck him and his prophet

Output: Not Hate Speech

Input: Literally the reason why I'm deleting my Instagram account. Sick of this fakery.

Output: Not Hate Speech

Input: Actually THIS WEIRD PEOPLE are HARMFUL! They need medications. His mother should have know he is like this and to not deny it

Output: Hate Speech

Input: You can ask me to call you a male, but you'll always be a woman

Output: Not Hate Speech

Input: I'm very sad about this fucking situation with all this muslims

Output: Not Hate Speech

Input: this is why indian subs should only count 0,5

Output: Hate Speech

Input: I remember once I was in charge of a disabled kid. Every night I was wishing slow death for him

Output: Hate Speech

Input: If your face was used on the inside of a jack-in-the-box there would be an epidemic of children dying from shock and disgust.

Output: Not Hate Speech

Input: This time the tables have turned

Output: Not Hate Speech

Input: Arrest that fat ass bitch!

Output: Hate Speech

Input: Trumps a liar and an idiot, a douche bag , Narcissist, and Zionist boot licker.

Output: Not Hate Speech

Input: I really like a lot of what you said here. I feel like i can barely speak up about this side of the topic because people are quick to shut down those who try to be too serious about it. 'Anxiety' and 'depression' and 'mental illness' have almost become buzz words. It seems that a lot of people don't know how difficult it is to really suffer and feel out of control of it all. As you spoke about, there's a fine line these days between awareness and influence. People get ideas in their heads and see the memes and the relateable content and start linking everything to their own lives. Before you know it, you've got perfectly healthy people being tainted by the world and people around them, imposing problems upon themselves and making life more difficult than it needs to be. It desensitises the whole situation and now I have people coming to me with real problems who don't want to speak up because of the upsurge in people talking about it. They feel they wouldn't be taken seriously. And that's horrible. I do understand though that it's an impossible seesaw to balance since so many people are involved and so many minds with a million ideas and actions are impossible to control and have on the same wave length.

Output:

**Answer:**

Not Hate Speech

### E.7    RTE

**Prompt:**

Input: At least 19 people have been killed in central Florida in the city of Lady Lake and Paisley after severe storms and a tornado ripped through the cities in the middle of the night. Eleven of those killed were in Paisley and three were in Lady Lake. The death toll is expected to rise as rescue crews resume tomorrow morning. Volusia, Sumter, Lake and Seminole counties have all been declared a state of an emergency as dozens of houses, mobile homes and a church were destroyed. Clothes and furniture are scattered around the wrecked houses and pieces of trees are scattered about. Cars are reported to have been turned over or thrown around in the air. "Our priority today is search and rescue," said Gov. of Florida, Charlie Crist. Rescuers are still looking through the wreckage to find survivors of those who might have been killed.

Gov. of Florida, Charlie Crist, has visited the cities of Lady Lake and Paisley.

Output: Does not entail

Input: Glue sniffing is most common among teenagers. They generally grow out of it once other drugs such as alcohol and cannabis become available to them. Seven-year-olds have been known to start "glue sniffing". Because of the social stigma attached to "glue sniffing" most snifters stop around 16 or 17 years, unless they are seriously addicted.

Glue-sniffing is common among youngsters.

Output: Entails

Input: Neil Armstrong was an aviator in the Navy and was chosen with the second group of astronauts in 1962. Made seven flights in the X-15 program (1960 photo), reaching an altitude of 207,500 feet. Was backup command pilot for Gemini 5, command pilot for Gemini 8, backup command pilot for Gemini 11, backup commander for Apollo 8, and commander for Apollo 11: successfully completing the first moonwalk.

Neil Armstrong was the first man who landed on the Moon.

Output: Entails

Input: Anna Politkovskaya was found shot dead on Saturday in a lift at her block of flats in the Russian capital, Moscow.

Anna Politkovskaya was murdered.

Output: Entails

Input: Argentina sought help from Britain on its privatization program and encouraged British investment.

Argentina sought UK expertise on privatization and agriculture.

Output: Does not entail

Input: The Security Council voted in 2002 to protect U.S. soldiers and personnel from other nations that haven't ratified the creation of the court through a treaty, and last June renewed the immunity for a year.

Immunity for soldiers renewed.

Output: Entails

Input: World leaders expressed concern on Thursday that North Korea will quit six-party nuclear disarmament talks and will bolster its nuclear weapons arsenal.

North Korea says it has a stockpile of nuclear weapons and is building more.

Output: Does not entail

Input: The Osaka World Trade Center is the tallest building in Western Japan.

The Osaka World Trade Center is the tallest building in Japan.

Output: Does not entail

Input: He endeared himself to artists by helping them in lean years and following their careers, said Henry Hopkins, chairman of UCLA's art department, director of the UCLA/Armand Hammer Museum and Cultural Center and former director of the Weisman foundation.

The UCLA/Hammer Museum is directed by Henry Hopkins.

Output: Entails

Input: Green cards are becoming more difficult to obtain.

Green card is now difficult to receive.

Output: Entails

Input: Nor is it clear whether any US support to Germany, in favour of Bonn as the WTO headquarters, would necessarily tilt a decision in that direction.

The WTO headquarters is in Bonn.

Output: Does not entail

Input: The Prime Minister's Office and the Foreign Office had earlier purposely asserted that the case is strictly in the jurisdiction of the police and the justice system.

The jurisdiction of the case was queried by the Prime Minister and the Ministry of Foreign Affairs.

Output: Does not entail

Input: Only a few Mag-lev trains have been used commercially such as at the Birmingham airport in the UK.

Maglev is commercially used.

Output: Entails

Input: Durham is the 'City of Medicine' and home of Duke University and North Carolina Central.

Duke University is in Durham.

Output: Entails

Input: Babe Ruth's career total would have been 1 higher had that rule not been in effect in the early part of his career. The all-time career record for home runs in Major League Baseball is 755, held by Hank Aaron since 1974.

Babe Ruth hit 755 home runs in his lifetime.

Output: Does not entail

Input: Boris Becker is a true legend in the sport of tennis. Aged just seventeen, he won Wimbledon for the first time and went on to become the most prolific tennis player.

Boris Becker is a Wimbledon champion.

Output: Entails

Input: Rabies is a viral disease of mammals and is transmitted primarily through bites. Annually, 7,000 to 8,000 rabid animals are detected in the United States, with more than 90 percent of the cases in wild animals.

Rabies is fatal in humans.

Output: Does not entail

Input: There are suppositions that the US Democratic Congress may re-establish the luxury taxes, which were already once introduced in the 1990s. The suppositions resulted in the National Association of Watch and Clock Collectors commissioning a report on various tax issues. Material goods such as jewelry, watches, expensive furs, jet planes, boats, yachts, and luxury cars had already been subjected to additional taxes back in 1990. After 3 years these taxes were repealed, though the luxury automobiles tax was still active for the next 13 years.

The US Congress may re-establish luxury taxes.

Output: Entails

Input: The U.S. handed power on June 30 to Iraq's interim government chosen by the United Nations and Paul Bremer, former governor of Iraq.

The U.S. chose Paul Bremer as new governor of Iraq.

Output: Does not entail

Input: FBI agent Denise Stemen said in an affidavit that Lowe's alerted the FBI recently that intruders had broken into its computer at company headquarters in North Carolina, altered its computer programs and illegally intercepted credit card transactions.

Non-authorized personnel illegally entered into computer networks.

Output: Entails

Input: A man who died during the G20 protests was pushed back by a police line minutes earlier, independent investigators have said. Ian Tomlinson, 47, who died of a heart attack, was blocked

from passing through a police cordon as he attempted to walk home from work at a newsagent, the Independent Police Complaints Commission (IPCC) said. He was caught on several CCTV cameras walking up King William Street where he was confronted by uniformed officers shortly before 7.30pm last Wednesday.

Ian Tomlinson was shot by a policeman.

Output: Does not entail

Input: GUS on Friday disposed of its remaining home shopping business and last non-UK retail operation with the 390m (265m) sale of the Dutch home shopping company, Wehkamp, to Industri Kapital, a private equity firm.

Wehkamp was based in the UK.

Output: Does not entail

Input: Shiite and Kurdish political leaders continued talks, on Monday, on forming a new government, saying they expected a full cabinet to be announced within a day or two.

US officials are concerned by the political vacuum and fear that it is feeding sectarian tensions, correspondents say.

Output: Does not entail

Input: San Salvador, Jan. 13, '90 (Acan-Efe) -The bodies of Hector Oqueli and Gilda Flores, who had been kidnapped yesterday, were found in Cuilapa, Guatemala, near the border with El Salvador, the relatives of one of the victims have reported.

Guatemala borders on El Salvador.

Output: Entails

Input: ECB spokeswoman, Regina Schueller, declined to comment on a report in Italy's la Repubblica newspaper that the ECB council will discuss Mr. Fazio's role in the takeover fight at its Sept. 15 meeting.

The ECB council meets on Sept. 15.

Output: Entails

Input: In June 1971 cosmonauts Georgi Dobrovolski, Vladislav Volkov, and Viktor Patsayev occupied Salyut for 23 days, setting a new record for the longest human spaceflight.

23 days is the record for the longest stay in space by a human.

Output: Entails

Input: The father of an Oxnard teenager accused of gunning down a gay classmate who was romantically attracted to him has been found dead, Ventura County authorities said today. Bill McInerney, 45, was found shortly before 8 a.m. in the living room of his Silver Strand home by a friend, said James Baroni, Ventura County's chief deputy medical examiner. The friend was supposed to drive him to a court hearing in his son's murder trial, Baroni said. McInerney's 15-year-old son, Brandon, is accused of murder and a hate crime in the Feb. 12, 2008, shooting death of classmate Lawrence "Larry" King, 15. The two boys had been sparring in the days before the killing, allegedly because Larry had expressed a romantic interest in Brandon.

Bill McInerney is accused of killing a gay teenager.

Output: Does not entail

Input: There is no way Marlowe could legally leave Italy, especially after an arrest warrant has been issued for him by the authorities. Assisted by Zaleshoff, he succeeds in making his escape from Milan.

Marlowe supported Zaleshoff.

Output: Does not entail

Input: A former federal health official arrested in the Virginia Fontaine Addictions Foundation scandal has been fined $107,000 for tax evasion. Patrick Nottingham, 57, was also sentenced to 18 months house arrest and ordered to complete 150 hours of community service work. The fine represents 50% of the federal income tax Nottingham did not pay on nearly $700,000 in kickbacks he received in return for approving excessive funding to the foundation in 1999 and 2000. In November 2005, Nottingham pleaded guilty to fraud and influence peddling and received a conditional sentence of two years less a day. "Mr. Nottingham was not only involved in fraudulent activity, he compounded that offence by not reporting that income," said Crown attorney Michael Foote at a sentencing hearing earlier this week. "He effectively committed two sets of extraordinarily serious offences." Nottingham's fine is the minimum allowed by law. Foote said there is little expectation Nottingham will ever pay off the fine.

Patrick Nottingham is involved in the Virginia Fontaine Addictions Foundation scandal.

Output: Entails

Input: Seoul City said Monday a 690-meter-tall, 133-story multifunctional skyscraper will be constructed in Sangam-dong. Once built, it will be the second highest after the 800-meter-high Burj Dubai, which is under construction, by South Korean developer Samsung C&T. The construction will cost more than 3.3 trillion won ($2.37 billion), the city estimates. To raise funds, 23 local developers signed an MOU at a Seoul hotel Monday with Seoul Mayor Oh Se-hoon attending. "The landmark building will help make Seoul more attractive and become a new tourist attraction here," Oh said. The multifunctional building will have hotels, offices, department stores, convention centers and various recreational facilities including an aquarium and movie theaters.

The highest skyscraper in the world is being built in Dubai.

Output: Entails

Input: Vodafone's share of net new subscribers in Japan has dwindled in recent months.

There have been many new subscribers to Vodafone in Japan in the past few months.

Output: Does not entail

Input: Swedish Foreign Minister murdered.

Swedish prime minister murdered.

Output: Does not entail

Input: Napkins, invitations and plain old paper cost more than they did a month ago.

The cost of paper is rising.

Output:

**Answer:**

Entails

E.8  LINEAR CLASSIFICATION

**Prompt:**

Input: 648, 626, 543, 103, 865, 910, 239, 665, 132, 40, 348, 479, 640, 913, 885, 456

Output: Bar

Input: 720, 813, 995, 103, 24, 94, 85, 349, 48, 113, 482, 208, 940, 644, 859, 494

Output: Foo

Input: 981, 847, 924, 687, 925, 244, 89, 861, 341, 986, 689, 936, 576, 377, 982, 258

Output: Bar

Input: 191, 85, 928, 807, 348, 738, 482, 564, 532, 550, 37, 380, 149, 138, 425, 155

Output: Foo

Input: 284, 361, 948, 307, 196, 979, 212, 981, 903, 193, 151, 154, 368, 527, 677, 32

Output: Bar

Input: 240, 910, 355, 37, 102, 623, 818, 476, 234, 538, 733, 713, 186, 1, 481, 504

Output: Foo

Input: 917, 948, 483, 44, 1, 72, 354, 962, 972, 693, 381, 511, 199, 980, 723, 412

Output: Bar

Input: 729, 960, 127, 474, 392, 384, 689, 266, 91, 420, 315, 958, 949, 643, 707, 407

Output: Bar

Input: 441, 987, 604, 248, 392, 164, 230, 791, 803, 978, 63, 700, 294, 576, 914, 393

Output: Bar

Input: 680, 841, 842, 496, 204, 985, 546, 275, 453, 835, 644, 1, 308, 5, 65, 160

Output: Bar

Input: 193, 101, 270, 957, 670, 407, 104, 23, 569, 708, 700, 395, 481, 105, 234, 785

Output: Foo

Input: 16, 409, 28, 668, 53, 342, 813, 181, 963, 728, 558, 420, 975, 686, 395, 931

Output: Bar

Input: 448, 421, 190, 246, 413, 766, 463, 332, 935, 911, 304, 244, 876, 95, 236, 695

Output: Foo

Input: 632, 318, 49, 138, 602, 508, 924, 227, 325, 767, 108, 254, 475, 298, 202, 989

Output: Foo

Input: 412, 140, 30, 508, 837, 707, 338, 669, 835, 177, 312, 800, 526, 298, 214, 259

Output: Foo

Input: 786, 587, 992, 890, 228, 851, 335, 265, 260, 84, 782, 33, 208, 48, 692, 489

Output: Foo

Input: 486, 76, 569, 219, 62, 911, 218, 450, 536, 648, 557, 600, 336, 17, 447, 838

Output: Foo

Input: 497, 654, 753, 787, 916, 672, 707, 121, 381, 867, 874, 725, 923, 739, 574, 612

Output: Bar

Input: 969, 665, 86, 219, 252, 723, 216, 918, 582, 401, 310, 408, 175, 91, 696, 266

Output: Foo

Input: 900, 609, 559, 506, 384, 265, 443, 466, 214, 526, 114, 17, 806, 666, 323, 65

Output: Foo

Input: 772, 104, 366, 321, 972, 345, 268, 760, 798, 70, 181, 170, 399, 313, 27, 85

Output: Foo

Input: 442, 799, 442, 461, 929, 258, 944, 533, 131, 16, 204, 593, 334, 492, 855, 477

Output: Foo

Input: 727, 176, 333, 15, 211, 614, 779, 757, 148, 635, 5, 423, 74, 383, 699, 162

Output: Foo

Input: 403, 586, 402, 130, 140, 260, 967, 916, 338, 293, 91, 371, 296, 735, 21, 683

Output: Foo

Input: 861, 487, 742, 886, 519, 263, 757, 918, 668, 425, 212, 169, 607, 647, 329, 788

Output: Bar

Input: 490, 968, 205, 971, 339, 13, 293, 226, 392, 331, 440, 670, 583, 219, 779, 928

Output: Foo

Input: 729, 140, 33, 748, 112, 179, 785, 257, 542, 815, 626, 248, 474, 821, 671, 654

Output: Bar

Input: 59, 874, 536, 60, 824, 223, 555, 809, 727, 448, 20, 482, 523, 928, 331, 182

Output: Bar

Input: 669, 414, 858, 114, 509, 393, 222, 627, 579, 336, 455, 732, 799, 636, 771, 990

Output: Bar

Input: 405, 146, 99, 760, 880, 778, 922, 555, 170, 600, 843, 358, 323, 654, 501, 603

Output: Bar

Input: 839, 45, 729, 900, 235, 605, 973, 304, 558, 479, 645, 77, 345, 768, 927, 734

Output: Bar

Input: 319, 605, 921, 13, 449, 608, 157, 718, 316, 409, 558, 364, 860, 215, 740, 909

Output: Bar

Input: 101, 969, 495, 149, 394, 964, 428, 946, 542, 814, 240, 467, 435, 987, 297, 466

Output:

**Answer:**

Bar

