# OpenReview forum: "Larger language models do in-context learning differently"
_ICLR.cc/2025/Conference — Submitted to ICLR 2025_

### Official Review · Reviewer_M5tP · 2024-10-16

**Soundness:** 3
**Presentation:** 3
**Contribution:** 3
**Rating:** 8
**Confidence:** 4

**Summary:**

The authors study the role of semantic priors in in-context learning. In order to do so, they study modified in-context learning settings in which the labels of the examples are flipped (flipped ICL) and in which the labels are replaced with semantically unrelated labels (SUL-ICL).

These experiments show that smaller models tend to stick to the same response as vanilla ICL when presented with flipped ICL examples. This shows that they ignore the concrete input-output examples in the prompt and instead stick to their semantic prior. In contrast, large models pick up the pattern present in the input-output examples and respond in the flipped way.

Similarly, they show that while small models don't benefit from an increased number of demonstrations in the SUL-ICL setting, larger models do.

Using the same settings they show that instruction tuned models stick more to their semantic priors in flipped ICL but also are better at picking up the pattern in the few shot examples in SUL-ICL.

**Strengths:**

The authors shed new light on in-context learning. Their modified ICL settings can help to disambiguate in-between looking up the right task vector (c.f. [1]) versus performing actual in-context learning. They refer to what I just called "looking up the right task vector" as sticking to the semantic prior.

Thereby, they demonstrate that larger models are required for actual ICL in the sense of incorporating the pattern present in the input-output examples into the LLMs repsonse.

The authors study many different model families across a variety of ICL tasks.


[1] https://arxiv.org/abs/2310.15213

**Weaknesses:**

The writing could be improved. E.g., the notion of semantic priors could be introduced earlier (maybe already in abstract) and in more detail (maybe in the introduction). The model names in the figure captions should be augmented with their respective size for easier readability for readers that don't know all the model name - model size combinations across the studied model families by heart.

The section about performing high-dimensional classification tasks via in-context learning somehow comes a bit out of the blue and is not nicely incorporated into the flow of the remaining paper.

As acknowledged by the authors in the limitation section the sample size of 100 per ICL task variant & model combination is rather small. Maybe to make sure the discovered trends are robust it would be worthwhile to do 1000 samples for one of the model families (as a reasonable trade-off between scientific insight and experimental cost).

**Questions:**

Do you think that studying the models of different sizes through the lens of mechanistic interpretability on the introduced ICL variations could provide valuable insight into the difference in behaviour in-between small and large models?

---

> ### Author Response · Authors · 2024-11-17
> **Author response to Reviewer M5tP**
>
> Thank you for the positive review and encouraging feedback. We are glad that the reviewer appreciated our extensive experimentation and insightful findings that shed new light on in-context learning.
>
> > Do you think that studying the models of different sizes through the lens of mechanistic interpretability on the introduced ICL variations could provide valuable insight into the difference in behaviour in-between small and large models?
>
> Thank you for this interesting suggestion! We agree that studying why these differing behaviors in in-context learning occur is an exciting research direction. While we view this experimentation as out of scope of this work because this work focuses on robustly demonstrating the existence of these differences, we are still excited by the possibility for future work to investigate this direction. As such, we’ve added the following paragraph to the revised version of our manuscript in the “Why are larger models better at in-context learning” subsection of the “frequently asked questions” section of the Appendix.
>
> “One way in which future work could gain insight into why larger models are better at in-context learning could be to study models using mechanistic interpretability. For example, analyzing attention and  activation  patterns  could  reveal  how  models  of  varying  sizes  process  in-context  examples.Furthermore, circuit analysis could identify specific mechanisms that might allow larger models to utilize new information over existing priors. Further analysis in this direction could potentially provide insight on how the ability to override priors is implemented and what structural changes enable stronger in-context learning capabilities.”

---

### Official Review · Reviewer_AN7L · 2024-10-20

**Soundness:** 4
**Presentation:** 4
**Contribution:** 2
**Rating:** 8
**Confidence:** 4

**Summary:**

The authors present a comprehensive study on the effect of semantic prior in In-Context Learning (ICL) settings. They use several classification tasks to measure ICL performance, designing four experiments:

1. Regular ICL
2. Flipped ICL: The authors flip a percentage of the labels in the context. They observe that larger models flip their answers more often. The authors conclude that larger models are less tied to their semantic priors.
3. Unrelated ICL: They swap the labels with semantically unrelated labels (e.g., yes/no --> foo/bar). They observe that larger models are more robust to this change. Surprisingly, instruct-tuned models are less robust to this change.
4. Linear classification: The authors evaluate the ICL performance of the models on high-dimensional (from 16 to 64) linear classification problems. Larger models perform better on these tasks.

**Strengths:**

- The authors present their work in a clear and concise manner.
- The experiments are well-designed, and the results are well-presented.
- The appendix is well-structured and provides additional information on the experiments.
- The experiments are comprehensive, spanning multiple datasets, architectures, and models.
- The studied phenomenon is interesting and highly relevant.y

**Weaknesses:**

The title is somewhat of an overstatement. The authors provide convincing evidence that larger models are better at ICL, but they do not provide evidence that these models do something fundamentally different from their smaller counterparts. A title like "Larger Language Models are Better In-Context Learners" or "Larger Language Models Are Less Tied to Semantic Priors" might be more appropriate.

While the authors perform a comprehensive study of the phenomenon (which is extremely valuable), they draw very few conclusions from the experiments. It would have been interesting to see a more in-depth analysis of WHY larger models are less tied to their semantic priors, or WHY instruct-tuned models become more tied to their semantic priors. However, the authors limit themselves to observing these phenomena without investigating their causes. To be fair, these additions may be slightly out of scope for the paper.

In section 6, the authors perform the ICL linear classification experiment. This is an interesting experiment, but most of the results are relegated to the appendix (Section D.4). Since the appendix section is relatively small, and the authors have another page to spare, I would suggest moving the results to the main paper. (Of course, the authors should prioritize the requests of other reviewers which may suggest more experiments.)

**Questions:**

This is evidently a very well-written and thoroughly investigated paper. My only reservation lies in the title, which I believe overstates the results of the paper. I am willing to increase my score if the authors agree to this small request.

I hope the authors find this review helpful. I am looking forward to reading the final version of the paper.

---

> ### Author Response · Authors · 2024-11-17
> **Author response to Reviewer AN7L**
>
> Thank you for the encouraging review - we appreciate your positive comments on the presentation of our results, extensiveness of our experiments, and relevance of our topic.
>
> We’ve revised the manuscript according to your feedback - please let us know if you have any further comments.
>
> > The title is somewhat of an overstatement. The authors provide convincing evidence that larger models are better at ICL, but they do not provide evidence that these models do something fundamentally different from their smaller counterparts.
>
> Thank you for this insightful comment on our manuscript title. We agree with your point and have revised the title to “Larger language models are better in-context learners” which we believe accurately reflects the phenomena that language models seem to be more capable at learning from in-context examples at larger scales. Note that while we are able to make this modification to the revised manuscript, the revision process does not allow us to change the title of the submission itself, only the title in the paper.
>
> > In section 6, the authors perform the ICL linear classification experiment. This is an interesting experiment, but most of the results are relegated to the appendix (Section D.4). Since the appendix section is relatively small, and the authors have another page to spare, I would suggest moving the results to the main paper.
>
> Thank you for pointing out this detail and we are glad that you found the experiment interesting. Given the available space, we agree that we can move the rest of the linear classification results to the main paper and have made this revision in the manuscript accordingly.
>
> > While the authors perform a comprehensive study of the phenomenon (which is extremely valuable), they draw very few conclusions from the experiments. It would have been interesting to see a more in-depth analysis of WHY larger models are less tied to their semantic priors, or WHY instruct-tuned models become more tied to their semantic priors. [...] To be fair, these additions may be slightly out of scope for the paper.
>
> Thank you for this important feedback on analyzing the causes behind the phenomena that we observe in our work. We agree with you that this is an important question to understand, but that it is slightly out of scope of our paper. We primarily view our paper as providing extensive and robust evidence that this phenomena **exists**, rather than attempting to also explain **why** it occurs. As such, we’ve included discussion on this in the “Why are larger models better at in-context learning” subsection of the “Frequently asked questions” section of the Appendix. We are excited to see future work further delve into analysis on why the behaviors shown in our work occur.

---

> > ### Comment · Reviewer_AN7L · 2024-11-17
> >
> > The authors addressed my concerns. I raised my score.

---

### Official Review · Reviewer_Cucr · 2024-11-04

**Soundness:** 4
**Presentation:** 4
**Contribution:** 2
**Rating:** 6
**Confidence:** 4

**Summary:**

This paper provides an analysis of the in-context learning abilities of models of different families, sizes, and training strategies (instruction-tuned vs non). Three ICL settings are explored: regular, flipped label, and semantically unrelated labels. Prior studies show mixed results regarding whether models primarily use prior semantic knowledge from pertaining or truly learn from inference-time examples. The current paper provides experimental results which show that smaller models rely more heavily on prior knowledge and are less capable of true ICL compared to larger models, and that instruction tuning seems to increase both ICL and the use of semantic priors.

**Strengths:**

The study provides a lot of data regarding their research questions. A wide variety of models and data are tested, and the analysis is both focused and rich. Some may say that the results are “obvious” given our experience of ICL, but the study seems to be thorough and if published will serve as evidence for arguments about ICL and scale which are currently made based on anecdotes.

**Weaknesses:**

Some in the ICLR community may find the results too obvious to justify giving them space at the conference.

**Questions:**

Is there any way to disentangle a model’s robustness to noisy input from reliance on semantic priors? The settings where greater than 0 but fewer than 100% of example labels are flipped might be interpreted in this way: The models learn the “true problem” they must solve from the regular examples and treat the flipped label examples as noise. One option would be to include an experiment comparing a model given N regular examples vs one given N regular and M flipped examples for various N and M.

---

> ### Author Response · Authors · 2024-11-17
> **Author response to Reviewer Cucr**
>
> Thank you for the thoughtful feedback. We are excited that the reviewer found our experiments to be extensive and to provide robust evidence for how in-context learning relates to model scale.
>
> > Is there any way to disentangle a model’s robustness to noisy input from reliance on semantic priors? The settings where greater than 0 but fewer than 100% of example labels are flipped might be interpreted in this way: The models learn the “true problem” they must solve from the regular examples and treat the flipped label examples as noise. One option would be to include an experiment comparing a model given N regular examples vs one given N regular and M flipped examples for various N and M.
>
> Thank you for this interesting comment. We agree that there is some subtlety in that flipping some labels may suggest to the model that the labels are noisy, rather than indicating to the model that it should start following those flipped labels. We believe, however, that this is still a function of reliance on semantic priors because models must decide when they no longer believe (a) that the original labels are correct and the inputs are just noisy and instead believe (b) that the original labels have actually been flipped, and the inputs are noisy in the other direction. This decision point is a function of the strength to which the model relies on its semantic priors, because a model that very heavily relies on its priors might treat a large number of flipped labels as “noise” before overriding its priors. This indicates to us that a model’s reliance on semantic priors directly affects its robustness to “noisy” input by influencing what the model considers “noise”.
>
> Please let us know if this helps clarify this nuance - we are happy to iterate more and are open to feedback/revising the paper if this is still unclear or if we’ve modeled this nuance incorrectly.

---

> > ### Comment · Reviewer_Cucr · 2024-11-24
> >
> > Thanks for your response.

---

### Official Review · Reviewer_5sFD · 2024-11-04

**Soundness:** 3
**Presentation:** 4
**Contribution:** 3
**Rating:** 6
**Confidence:** 4

**Summary:**

The study investigates the effects of semantic priors versus input-label mappings on in-context learning (ICL) in language models, exploring two scenarios: ICL with flipped labels and ICL with semantically unrelated labels. The findings reveal that large language models demonstrate an emergent ability to override semantic priors when presented with contradicting in-context examples, a capability that smaller models lack. Additionally, the study introduces semantically-unrelated label ICL (SUL-ICL), where models must learn input-label mappings from in-context examples. This ability also emerges with model scale, with larger models capable of performing linear classification in SUL-ICL settings. Finally, evaluation of instruction-tuned models indicates that instruction tuning enhances the reliance on semantic priors and improves the ability to learn input-label mappings, though it primarily strengthens the former.

**Strengths:**

1.	This paper is well-written. The major analysis point is clear and the experimental design makes sense to me. A large number of experiments are conducted in this paper to support the point that the ability to override the semantic prior trained during the pretraining process and learn new input-label mappings from the context emerges with larger scales.

2.	The analysis part about instruction-tuned models that they are worse at overriding the semantic priors is quite surprising and could inform people about the two-side effect of the instruction tuning.

**Weaknesses:**

1.	As mentioned in the limitations, more experiments on the generation tasks could make this paper stronger. One possible way would be inserting wrong/different facts from the semantic prior and see if the model would respond based on the newly inserted facts.

2.	I’m curious if the conclusions about the sizes would still stand in the newly trained LLMs with better structures and training corpus. It is difficult to tell if this behavior is really directly related to the model sizes or if it could be also affected/changed by other factors/techniques.  Some results on the Llama3, Mistral, or Qwen models could better prove the authors’ point.

3.	I am still not fully convinced about the practical value of the findings in this paper even after reading the FAQs in the appendix part. Considering the behavior of semantic prior overriding could be both beneficial (manipulate existing knowledge) and harmful (noise or false knowledge in the content), I think a section discussing how to control the behavior of overriding the semantic priors generally (or separately for small and large language models) could make this paper stronger by providing more controllability to the current LLMs.

**Questions:**

1.	Could the behavior of overriding semantic prior be acquired not only by using larger models but also by using better training techniques and corpus?

2.	Are there any methods to control the semantic-prior overriding behavior in different cases (e.g., could we make larger models perform less in this way or vice versa? )? The practical value would be largely decreased if only larger or smaller language models could behave in one specific way.

---

> ### Author Response · Authors · 2024-11-17
> **Author response to Reviewer 5sFD (1/2)**
>
> Thank you for these insightful questions. We are happy to see that you appreciated our extensive experimentation and analysis on instruction-tuned models.
>
> We revised our manuscript and added discussion following your suggestions. Please feel free to let us know if you have additional questions!
>
> > Could the behavior of overriding semantic prior be acquired not only by using larger models but also by using better training techniques and corpus? I’m curious if the conclusions about the sizes would still stand in the newly trained LLMs with better structures and training corpus. [...]
>
> Thank you for this important question. In our work, we investigated several model families to study how in-context learning behaviors differ across model scale. Namely, each model in the LLM model family is trained on the same training data and uses the same training protocol; the only difference is model size.  This means that our findings on the LLM and IT-LLM models show the effect of purely scaling model parameter count.On the other hand, we also used GPT-3, InstructGPT, and Codex models; for these families, it is not publicly known if the models within each family are only scaling in terms of parameter count. It is likely that there are other scaling factors (e.g., better training data, a better model architecture) that increase the scale of that model. For example, code-davinci-001 and code-davinci-002 may actually have the same number of parameters, but code-davinci-002 could be trained on better data. In our paper, we assume that for these model families, the models are increasing in scale for some scaling factor, not just parameter count.
>
> Our experiments show that in-context learning behavior often changes with respect to model scale for all model families. This means that (a) purely scaling parameter count can result in the behavior of overriding semantic priors and (b) models that are larger for scaling factors other than parametercount (e.g., quality of training data) still exhibit the behavior of being able to override semantic priors when performing in-context learning. We can see (a) because the behavior exists in the LLM model family, for which models only differ by parameter count. We can see (b) because the behavior exists on the GPT-3, InstructGPT, and Codex model families, for which models do not necessarily only differ by parameter count (although we cannot confirm which exact factors are contributing because this information is not publicly known).
>
> We hope this helps answer your question. We’ve added this discussion in the “Frequently Asked Questions” section of the Appendix in the revised manuscript.
>
> > Are there any methods to control the semantic-prior overriding behavior in different cases (e.g., could we make larger models perform less in this way or vice versa? )? [...] I think a section discussing how to control the behavior of overriding the semantic priors [...] could make this paper stronger [...].
>
> Thanks for this insightful question on how one might be able to control the behavior of overriding semantic priors when performing in-context learning. We agree that this is an important point of discussion, and as such, we have added the following discussion into a new subsection titled “How can one control in-context learning behaviors?” in the “Frequently-asked questions” section of the Appendix.
>
> “The behavior of overriding semantic priors can be both beneficial (e.g., teaching the model an updated fact) and harmful (leaving the model susceptible to false knowledge in a prompt). An open question is thus how one might be able to control the behavior of overriding semantic priors in language models.
>
> One example of a harmful side effect of language model’s susceptibility to override priors when presented with in-context examples is many-shot jailbreaking (Anil et al., 2024), where a large number of in-context examples of unsafe dialogues are presented to a language model in order to override its prior safety training. Anil et al. (2024) found that a useful intervention to prevent this strategy of jailbreaking is to finetune the language model on examples where the model follows its prior knowledge and ignores demonstrations given in-context.  These findings suggest that some control over the extent to which large language models override priors with in-context examples can be gained via supervised finetuning.  For example, if one desires a large language model that always conforms to its prior knowledge, supervised finetuning on examples where the final answer is independent of the few-shot examples may reduce the model’s tendency to override prior knowledge when shown in-context examples.”

---

> ### Author Response · Authors · 2024-11-17
> **Author response to Reviewer 5sFD (2/2)**
>
> > As mentioned in the limitations, more experiments on the generation tasks could make this paper stronger. One possible way would be inserting wrong/different facts from the semantic prior and see if the model would respond based on the newly inserted facts.
>
> Thanks for this neat suggestion! While we view generative tasks as out of scope for our findings because of the difficulties with evaluating the model’s answer in a generative setting and whether this type of injection of facts would work for non-instruction-tuned models (which may not understand that they are supposed to try to learn from the facts). As such, we’ve added your experimental suggestion into our existing “Would these findings translate to generative tasks” subsection of the “Frequently asked questions” section in the Appendix.

---

> ### Comment · Reviewer_5sFD · 2024-11-25
>
> Thanks for the response! They solve most of my concerns and I'll keep my current score.

---

### Official Review · Reviewer_J5hP · 2024-11-04

**Soundness:** 3
**Presentation:** 2
**Contribution:** 2
**Rating:** 1
**Confidence:** 5

**Summary:**

The following paper analyzes the role of ICL techniques in LLMs when influenced by semantic heuristics. The authors anialkize We analyzed two setups -- ICL with inverted labels and ICL with semantically unrelated labels -- in different model families. The major findings point out that there are spiccatr abilities beyond semantics in models with more paramenters while. So the authors also study instruction-tuned models by observing that semantic heuristics can have effects even in low-scale models.

**Strengths:**

The paper is very interesting and argues hot topics however although the experiments are exhaustive they are not best introduced and could be discussed in more detail.

**Weaknesses:**

The paper does not expose scientific novelty.

Although the experiments and the (slightly crude) discussion are good, these experiments or something similar has been presented here before: https://neurips.cc/virtual/2023/76728.

I would be grateful to the authors if they could highlight the new paper's substantial improvements.

**Questions:**

Please read the Weaknesses

---

> ### Author Response · Authors · 2024-11-17
> **Author response to Reviewer J5hP**
>
> Thank you for this important review - we are glad that you found our paper interesting and appreciated our extensive experimentation.
>
> We revised the paper based on the feedback you gave. Please let us know if you have any further comments or suggestions.
>
> > The paper does not expose scientific novelty. Although the experiments and the (slightly crude) discussion are good, these experiments or something similar has been presented here before: https://neurips.cc/virtual/2023/76728.I would be grateful to the authors if they could highlight the new paper's substantial improvements.
>
> Thanks for bringing up this related work. We are happy to modify our paper to discuss this work and how it relates to our own findings. In our revision, we’ve added discussion on this work’s findings into our related work section. We believe that there is an important distinction between the work from Shi et al. and our findings; while both our work and Shi et al. examine the behavior of how larger language models perform in-context learning differently, our work focuses on **empirically** demonstrating this phenomenon across many task setups and model families, whereas Shi et al. focuses on theorizing about **why** this phenomenon occurs. Because our findings robustly demonstrate that this behavior **occurs** in the first place and Shi et al. provide explanations for **why** this behavior occurs, we are confident that our findings provide substantial novelty with respect to Shi et al.’s work.
>
> Here’s our addition to the “Related Work” section in the revised manuscript. We’ve aimed to reflect the important distinction between empirical demonstration of a behavior and theoretical explanations for why the behavior occurs. We welcome any feedback on how to best frame these differences and are open to suggestions.
>
> “Our work makes similar claims about the ability for language models to learn tasks via input--label mappings only, though it differs crucially in that we observe frozen pretrained transformers without any additional learning. Additionally, our work focuses on empirically demonstrating that larger language models are better at learning input--label mappings in-context. Related work from Shi et al. (2024) provides theoretical explanations for this phenomenon, proposing that when performing in-context learning, larger language models cover more hidden features, whereas smaller language models emphasize important hidden features.”
>
> We’ve also added a reference to this work in the “Why are larger language models better at in-context learning” subsection of the “Frequently asked questions” section of the Appendix in the revised manuscript.

---

> > ### Comment · Reviewer_J5hP · 2024-11-25
> > **Answer**
> >
> > Thank you for the detailed answer however are you claiming that the previous work is an ancestor of yours? If so your analysis should really provide huge detail. Please could you list them in detail as it is not easy to observe them. Thank you in advance for your reply.

---

> > > ### Author Response · Authors · 2024-11-25
> > >
> > > Thanks for the reply. As stated in our response and in the revised version of the paper, we believe that our work and Shi et al.'s findings are substantially different because our experiments demonstrate that the behavior where larger language models perform in-context learning differently **exists**, whereas Shi et al. attempts to explain **why** this behavior occurs. We are not entirely sure what is meant by "ancestor" - we believe that this work provides significant novelty beyond Shi et al.'s findings.
> > >
> > > We would love any further clarifications on what you mean by "ancestor" and what precise details you believe were not adequately discussed. We would be happy to address any concrete and specific feedback you may have and revise the manuscript accordingly.

---

### Meta-Review · Area_Chair_an1U · 2024-12-18

**Metareview:**

This paper studies in-context learning with language models, in particular how a model's reliance on "semantic priors" vs. the input-output mappings given in context is affected by model size. It finds that larger models are better able to leverage the input-output mappings given in context.

On the plus side, this paper empirically studies an important phenomenon. On the negative side, the findings are not surprising, the experiments are limited in scope (for 2024), and the writing could be improved. Had this paper come out shortly after Min et al.  2022 ("Rethinking the Role of Demonstration..."), I may have been more excited about it, but at this point, I can't see myself getting excited about this paper.

**Additional Comments On Reviewer Discussion:**

Reviewer J5hP gave a very low score given similarities to prior work, and did not change the score after the author rebuttal. While I am somewhat sympathetic to the authors in the perhaps overly harsh, I also agree with this reviewer overall that this is quite similar to existing work.

Reviewer AN7L raised their score in response to the author rebuttal. I did not find the author rebuttal to meaningfully change the (nontrivial) weaknesses highlighted by this reviewer.

---

### Decision · Program_Chairs · 2025-01-22

Reject